# F-actin architecture determines the conversion of chemical energy into mechanical work

Ryota Sakamoto [1,2] & Michael P. Murrell [1,2,3] ✉

Mechanical work serves as the foundation for dynamic cellular processes, ranging from cell division to migration. A fundamental driver of cellular mechanical work is the actin cytoskeleton, composed of filamentous actin (F-actin) and myosin motors, where force generation relies on adenosine tri-phosphate (ATP) hydrolysis. F-actin architectures, whether bundled by cross-linkers or branched via nucleators, have emerged as pivotal regulators of myosin II force generation. However, it remains unclear how distinct F-actin architectures impact the conversion of chemical energy to mechanical work. Here, we employ in vitro reconstitution of distinct F-actin architectures with purified components to investigate their influence on myosin ATP hydrolysis (consumption). We find that F-actin bundles composed of mixed polarity F-actin hinder network contraction compared to non-crosslinked network and dramatically decelerate ATP consumption rates. Conversely, linear-nucleated networks allow network contraction despite reducing ATP consumption rates. Surprisingly, branched-nucleated networks facilitate high ATP consumption without significant network contraction, suggesting that the branched network dissipates energy without performing work. This study establishes a link between F-actin architecture and myosin energy consumption, elucidating the energetic principles underlying F-actin structure formation and the performance of mechanical work.

Essential biological functions, from morphogenesis to wound healing, are carried out through the mechanical work of cells[1–3]. At the single-cell level, mechanical behaviors, such as cell division and migration, are powered by the hydrolysis of adenosine triphosphate (ATP) into adenosine diphosphate (ADP), releasing the high free energy stored in the phosphate bond[4]. The actin cytoskeleton plays pivotal roles in driving cellular dynamics, consisting of actin filaments and myosin II motors. Actin filaments (F-actin) are in a non-equilibrium dynamic steady-state, continuously undergoing assembly and disassembly of actin monomers (G-actin), termed turnover, which is sustained by ATP hydrolysis (consumption)[5]. When ATP-bound myosin binds to F-actin, ATP hydrolysis induces a power stroke on F-actin, generating a

contractile force within the actomyosin network[6]. Thus, the continuous energy supplied by ATP hydrolysis is essential for performing mechanical work and driving various cellular behaviors. However, not all the chemical energy released from ATP hydrolysis is converted into mechanical work, limiting energy conversion efficiency (Fig. 1a).

As motors and filaments assemble into large-scale architectures, there are numerous steps in which mechanical work is generated. However, the role of F-actin architecture on the molecular consumption of energy in cell-scale production of work is unknown. To date, the role of F-actin architecture is appreciated in the generation and transmission of force[7–15]. Recent studies have reported that crosslinked F-actin bundles formed by actin-crosslinkers, such as fascin (which

[1]Department of Biomedical Engineering, Yale University, 10 Hillhouse Avenue, New Haven, CT, USA. [2]Systems Biology Institute, 850 West Campus Drive, West Haven, CT, USA. [3]Department of Physics, Yale University, 217 Prospect Street, New Haven, CT, USA. ✉e-mail: michael.murrell@yale.edu

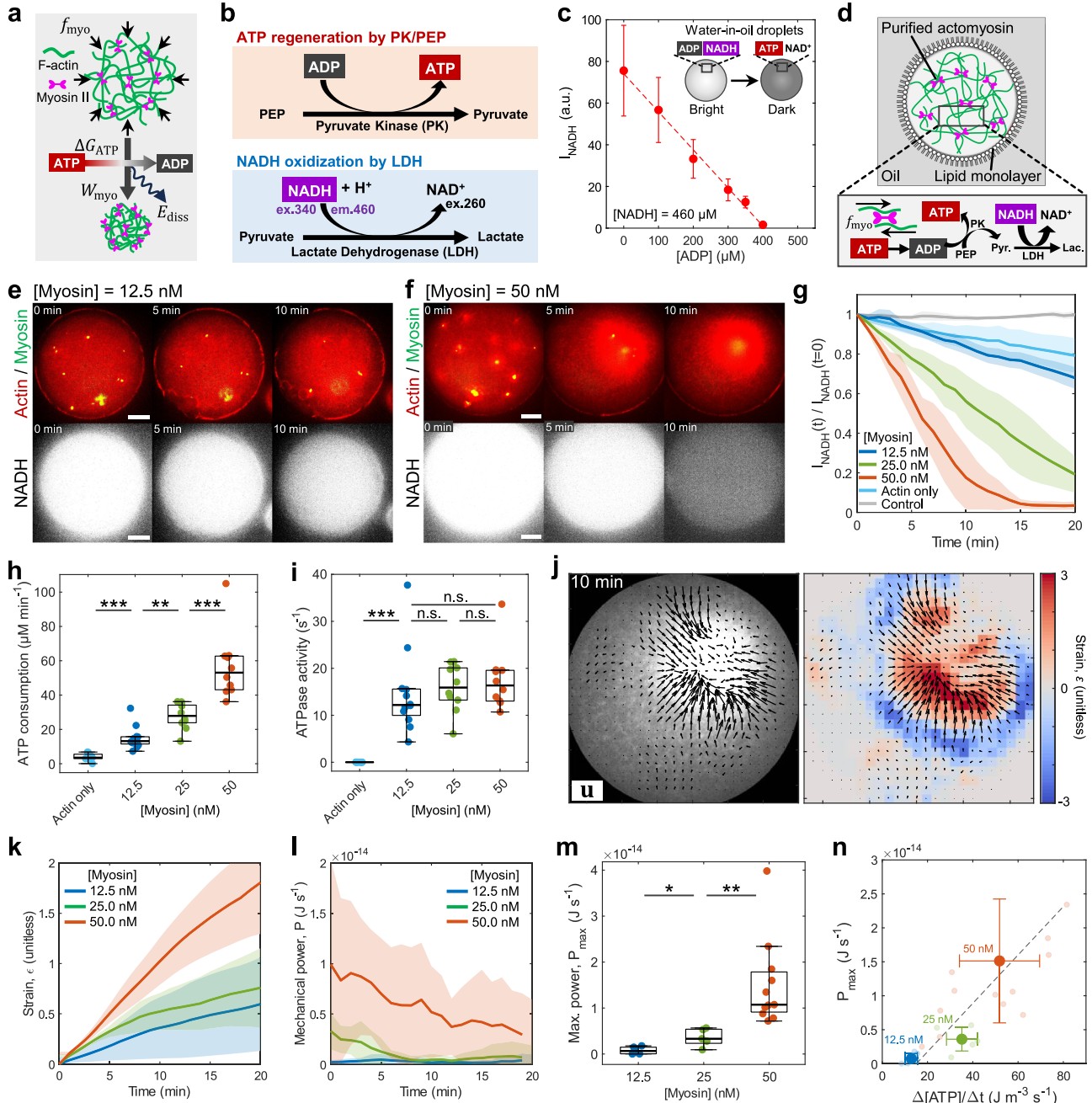

**Fig. 1 | ATP consumption rate measurement using NADH-coupled assay and mechanical power estimation. a** Schematic of the free energy released upon ATP hydrolysis ($\Delta G_{ATP}$) converted to the mechanical work performed by myosin on actin network contraction ($W_{myo}$). $E_{diss}$ is the dissipated energy not used for work. $f_{myo}$ is the force generated by single motors. **b** Schematic of the NADH-coupled assay. **c** NADH fluorescence dependence on ADP concentration in the NADH-coupled assay. The initial NADH concentration is fixed at 460 μM ($n = 466$ droplets and $N = 8$ independent experiments). **d** Schematic showing actomyosin-encapsulated droplets coupled to the NADH assay. **e, f** Time-lapse images showing the contraction of actomyosin with myosin concentration at 12.5 nM (e) and 50 nM (f) (actin in red, myosin in green) and the NADH fluorescence (in white). **g** NADH fluorescence over time normalized by the initial time point ($n = 7$ and $N = 4$ in control only containing NADH; $n = 11$ and $N = 3$ for actin only; $n = 8$ and $N = 3$ for 12.5 nM; $n = 7$ and $N = 3$ for 25 nM; $n = 9$ and $N = 3$ for 50 nM). **h** Boxplot showing the ATP consumption rate in (**g**) ($n = 11$ and $N = 3$ for actin only; $n = 12$ and $N = 4$ for 12.5 nM; $n = 10$ and $N = 4$ for 25 nM; $n = 10$ and $N = 4$ for 50 nM). **i** ATPase activity calculated by the ATP consumption rate in (**h**) divided by the concentration of actin or myosin ($n = 11$ and $N = 3$ for actin only; $n = 12$ and $N = 4$ for 12.5 nM; $n = 10$ and

$N = 4$ for 25 nM; $n = 10$ and $N = 4$ for 50 nM). **j** Actomyosin network contraction (left) with 50 nM myosin. Black arrows are the total displacement over 10 min and vector magnitudes are normalized by its maximum. Colormap represents local strain fields (right). **k** Mean compressive strain of the actin network over time ($n = 6$ and $N = 3$ for 12.5 nM; $n = 5$ and $N = 3$ for 25 nM; $n = 10$ and $N = 9$ for 50 nM). **l** Instantaneous power of the actin network over time ($n = 6$ and $N = 3$ for 12.5 nM; $n = 5$ and $N = 3$ for 25 nM; $n = 12$ and $N = 10$ for 50 nM). **m** Maximum instantaneous power performed by myosin extracted from (**l**) ($n = 6$ and $N = 3$ for 12.5 nM; $n = 5$ and $N = 3$ for 25 nM; $n = 11$ and $N = 10$ for 50 nM). **n** Scatter plot showing the maximum power dependence on ATP consumption rate ($n = 6$ and $N = 3$ for 12.5 nM; $n = 5$ and $N = 3$ for 25 nM; $n = 11$ and $N = 10$ for 50 nM). Individual data is shown as small markers. The dashed line is the linear fitting to the individual data. Data are presented as mean ± SD (**c, g, k, l, n**) or boxplots where the interquartile range (IQR) is between Q1 (25th percentile) and Q3 (75th percentile), the center line indicates the median, whiskers are extended to Q3 + 1.5 × IQR and Q1 − 1.5 × IQR (**h, i, m**); *$p < 0.05$; **$p < 0.01$; ***$p < 0.001$ in a two-sided Wilcoxon rank sum test. n.s., not significant. Scale bars, 10 μm. Source data are provided as a Source Data file.

aligns the polarity of F-actin in the same direction) or fimbrin (which mixes the polarity of F-actin), alter the walking behavior of myosin II thick filaments bound to a 2-dimensional F-actin network[16]. Furthermore, it has been reported that linear actin networks nucleated by formin mDia1 can be contracted by myosin, whereas branched actin networks nucleated by the Arp2/3 complex impede myosin II contraction in a 2-dimensional actomyosin network[11]. In this case, it has been argued that the dense, branched actin gels attenuate the contractile force of myosin II. Thus, F-actin architecture, mediated by crosslinkers and nucleators, is crucial in controlling the contractile behaviors of myosin. In living cells, distinct F-actin architectures are observed depending on cell function. For example, stress fibers in adherent cells and contractile rings during cell division are composed of linear F-actin nucleated by formin[17,18], while branched F-actin structures are observed in retrograde flow during cell migration[19,20]. Therefore, F-actin architecture-dependent myosin behaviors could potentially play pivotal roles in controlling cellular force generation.

However, it remains unclear how F-actin architecture-dependent contractile behaviors and energy consumption are interconnected. For example, although both mixed polar bundled networks and branched actin networks restrict myosin contractility, it remains unknown whether these structures can conserve energy by inhibiting myosin ATP hydrolysis or they still allow myosin energy consumption while decreasing the performance of mechanical work. Since distinct F-actin architectures underlie the foundation of essential cellular processes, understanding the regulation of myosin force generation and energy consumption by F-actin architecture could enable us to control cellular states based on energetic principles.

In cells, quantifying how much ATP consumption originates from myosin ATP consumption is currently infeasible, as many other cellular processes utilize ATP hydrolysis, such as protein synthesis. Furthermore, the coexistence of distinct F-actin architectures poses challenges to measuring F-actin architecture-dependent myosin energy consumption. To address these challenges, we reconstitute various F-actin architectures in vitro[7,13–15,21–26], and measure the ATP consumption rate of myosin using the nicotinamide adenine dinucleotide (NADH)-coupled assay[27,28]. We find that ATP hydrolysis depends upon the organization of actomyosin. First, we find that polarity of actin bundles differentially regulates ATP consumption rates, with mixed polar bundles significantly reducing both ATP consumption and mechanical work. On the other hand, linear actin networks reduce ATP consumption but allow for network contraction. In contrast, branched actin networks consume ATP as rapidly as spontaneously polymerized actin while performing minimal mechanical work. These results emphasize the significance of F-actin architecture not only in force generation but also in energy consumption rates and in the efficiency of energy conversion. This finding may have implications for understanding how cellular structure formation is autonomously controlled based on the energetic principles of F-actin architecture-dependent performance of mechanical work and energy consumption by myosin.

## Results

### ATP consumption rate during actomyosin contraction measured by the NADH-coupled assay

Purified actin and skeletal muscle myosin II are mixed with F-actin nucleators and crosslinkers, allowing for precise control of distinct F-actin architectures[11,16,29,30]. Simultaneously, ATP consumption rate of myosin is measured by the NADH-coupled assay, where NADH fluorescence serves as a readout of the amount of consumed ATP (Fig. 1b, Supplementary Fig. 1)[27,28]. In the presence of ADP, ATP regeneration system produces the byproduct pyruvate, which is subsequently transformed into lactate, accompanied by the oxidization of NADH to $NAD^+$. Since NADH fluorescence is excited at 340 nm, while $NAD^+$ absorbance is at 260 nm[31], this oxidization diminishes the fluorescence of NADH, and the level of diminished fluorescence corresponds to the

amount of ADP converted to ATP, owing to the 1:1 stoichiometric ratio of ADP to NADH (Fig. 1c). Therefore, when the NADH-assay is coupled to an actomyosin system, the fluorescence of the droplets is reduced as more ATP is consumed by myosin, where the slope of the NADH fluorescence decay corresponds to the ATP consumption rates.

Encapsulation of these two systems, purified actomyosin and the NADH-coupled assay, within water-in-oil droplets covered by a lipid monolayer allows us to quantify both the mechanical work performed by myosin and the overall ATP consumption rates (Fig. 1d). The water-in-oil droplets serve as a pico-liter non-deformable closed chamber suited for measuring ATP consumption rates of enclosed biological systems[27] (Method). Actomyosin-in-oil droplets were prepared on ice. Immediately after emulsification, emulsion droplets were transferred onto a glass coverslip, thereby the temperature was raised to R.T. (25 °C) to initiate actin polymerization and myosin contraction. Approximately 10 s after increasing the temperature, the actomyosin network initiated contraction (Fig. 1e, f, Movie S1 and S2).

Firstly, we demonstrate that a higher myosin concentration accelerates the ATP consumption rates ($\Delta[ATP]/\Delta t = 15.0 \pm 6.6\,\mu M\,min^{-1}$, mean $\pm$ std, at 12.5 nM; $\Delta[ATP]/\Delta t = 56.6 \pm 19.4\,\mu M\,min^{-1}$ at 50 nM), establishing the experimental setup for simultaneously imaging the dynamics of actomyosin contraction and quantifying the total amount of consumed ATP (Fig. 1g–i). Using the value of free energy released from ATP hydrolysis in a physiological condition ~50–70 kJ $mol^{-1}$[32,33], and a typical droplet radius $R = 25\,\mu m$, we estimate the total free energy consumed in a droplet to be ~$3.7 \times 10^{-12}$ J $s^{-1}$. We analyzed the comparable sizes of droplets with diameter ~45–65 μm to reduce the variability (Supplementary Fig. 2). The background ATP hydrolysis rate of actin is lower than the myosin ATP consumption rate ($\Delta[ATP]/\Delta t = 4.0 \pm 2.0\,\mu M\,min^{-1}$ at 6 μM actin) (Fig. 1h, Supplementary Fig. 3). We confirmed that the inhibition of the ATPase activity of myosin by s-nitro-blebbistatin[34] significantly slows down the ATP consumption rate ($\Delta[ATP]/\Delta t = 8.7 \pm 3.6\,\mu M\,min^{-1}$ with 1 mM s-nitro-blebbistatin at 50 nM myosin) (Supplementary Fig. 4, Movie S3). The ATP consumption rate of myosin in the absence of F-actin was less than 10% of the myosin ATPase activity activated by F-actin ($\Delta[ATP]/\Delta t = 5.5 \pm 1.8\,\mu M\,min^{-1}$ with 50 nM myosin only) (Supplementary Fig. 5). The ATPase activity of the individual myosin motors, estimated from the ATP consumption rate divided by myosin concentration, was found to be comparable across different myosin concentrations ($14.6 \pm 8.8\,s^{-1}$ at 12.5 nM; $15.9 \pm 5.0\,s^{-1}$ at 25 nM; $17.5 \pm 6.5\,s^{-1}$ at 50 nM) (Fig. 1i). This value is comparable to the ATPase activity of skeletal muscle myosin II measured by the radiolabeled assay, ~20 $s^{-1}$[35].

### Estimation of the lower bound mechanical work performed by myosin during network contraction

The mechanical work is estimated based on the changes in the volume of the F-actin network. Briefly, using Particle Image Velocimetry (PIV), we measure displacement vectors between individual frames (1 min interval) at every spatial location to calculate local strain field (Fig. 1j) (Methods). The cross-sectional area of the contracting actin network is approximated as a circle with radius $R$. This approximated radius of the actin network is used to calculate the volume of the network, $V \sim (4/3)\pi R^3$, the instantaneous mechanical work performed by myosin, $dW_{myo} = -\sigma_{myo}dV$, where $\sigma_{myo}$ is the active stress generated by myosin, and the instantaneous power, $P = dW_{myo}/dt$. It should be noted that this is the lower bound estimation, since the bending energy of F-actin[29] and non-linearity of actomyosin contraction is not taken into account[36]. We confirmed that the PIV-based mechanical power estimation is comparable to the mechanical power independently calculated by the actin fluorescence-based volume change estimation method (Supplementary Fig. 6). The active stress induced by myosin acting on the actin network is estimated using the total myosin-generated force within a droplet: $F_{myo}^{tot} \sim f_{myo}c_{myo}V_0d$, where $f_{myo} = 3.4$ pN is the force generated by single motors[37], $c_{myo}$ is the

myosin concentration, $V_0$ is the initial network volume, and $d = 0.05$ is the duty ratio of skeletal muscle myosin II[38]. For simplicity, we assumed (i) myosin motors act independently, and (ii) there is no load-dependence of myosin motors. The myosin active stress is defined as $\sigma_{myo} = F^{tot}_{myo}/(4\pi R^2)$ and is used for the calculation of mechanical work. It is important to note that the mechanical power estimates in this study are based on changes in the actin network volume (e.g., pressure-volume work). Consequently, there may be additional contributions to the mechanical work performed by myosin, such as (i) volume conservation modes like filament buckling[30,39,40] and stretching[41–44], (ii) dissipative effects like filament breakage[30,45–47], and (iii) load/architecture-dependent myosin force generation[11,16]. Thus, our estimates serve as a lower bound. However, we demonstrate that the work of pressure-volume changes significantly exceeds those increased by the additional contributions (Supplementary Notes 1 and 2).

During the contraction of the actomyosin network, the higher myosin concentration induced faster contraction with higher power (Fig. 1k, l). The maximum power performed by myosin during contraction increased for the higher myosin concentration[36] ($P_{max} = (7.7 \pm 7.5) \times 10^{-16}$ J s$^{-1}$ at 12.5 nM; $P_{max} = (1.5 \pm 1.0) \times 10^{-14}$ J s$^{-1}$ at 50 nM) (Fig. 1m). We find that the larger energy input (ATP consumption rate, $\Delta[ATP]/\Delta t$) results in the greater mechanical output ($P_{max}$) (Fig. 1n). This suggests the influence of the load-dependence of ATP hydrolysis of myosin[48,49] may not be significant during the contraction of the spontaneously polymerized actin network in the present setup. We confirmed that the reduced ATP concentration resulted in a slightly decelerated ATP consumption rate and mechanical power (Supplementary Fig. 7, Movie S4), which aligns with reports indicating that the myosin II sliding velocity in muscle fibers and in in vitro motility assays increases proportionally to ATP concentration[50,51]. At low ATP concentration, the network may exhibit increased transient crosslinking due to the higher affinity of ADP-myosin for F-actin[52], which could also contribute to load-dependent ATP hydrolysis.

The energy conversion efficiency was estimated from the ratio of mechanical power ($P_{max}$) to the rate of free energy released from ATP hydrolysis ($\Delta G_{ATP}/\Delta t$) to be ~0.0041 at 50 nM myosin (Supplementary Fig. 8). Notably, this value is ~100 times smaller than the maximum efficiency estimated for the contraction of skeletal muscle fiber (~0.36[53,54]), highlighting the importance of the structural organization of the actomyosin system to efficiently extract macro-scale work from the molecular consumption of energy. We found that the efficiency increases with higher myosin concentration, perhaps because the elevated myosin concentration may increase the probability of temporary connections within the actin network via myosin thick filaments, improving the transmission of contractile forces[55,56] (Supplementary Fig. 8). Having established that a higher myosin concentration increases both ATP consumption rate and mechanical power in a spontaneously polymerized actin network, we next explore the impact of F-actin architecture on myosin-based mechanical power and energy consumption rates.

## The mixed polarity bundles hinder network contraction and decelerate ATP consumption rates

In the following experiments, we maintain the myosin concentration at 50 nM while altering the F-actin architecture to assess its impact on mechanical work and ATP consumption rate. First, we examine the influence of F-actin polarity within bundles. Since myosin walks towards the barbed end of F-actin, the polarity of the F-actin plays pivotal roles in contractile behaviors[7,16]. To investigate how the polarity of the F-actin network can influence myosin ATP consumption, we utilized two types of crosslinkers, fascin and fimbrin (Fig. 2a). Fascin is a small (~6 nm) actin crosslinking protein that forms F-actin bundles with filaments aligned in the same polarity (i.e., unipolar)[57]. In contrast, fimbrin is a small protein (~7 nm) that forms F-actin bundles where the

polarity of F-actin is randomly oriented (i.e., mixed polar)[58]. We chose sufficiently high concentration of crosslinkers to impact myosin force generation as has been established in previous studies[16].

Notably, we observed that unipolar bundles crosslinked by fascin (1 μM fascin; [Actin]:[Fascin] = 1:0.17) contracted as rapidly as non-crosslinked network, while mixed polar bundles crosslinked by fimbrin (1 μM fimbrin; [Actin]:[Fimbrin] = 1:0.17) did not contract, and the F-actin network was distributed across the droplets (Fig. 2b–e, Movie S5 and S6). The homogeneous structure of strain fields in fascin corresponds to the uniform contraction of the entire network, as observed in spontaneously polymerized networks (Fig. 2d). The mechanical power performed by non-crosslinked actin network and unipolar bundles with fascin are comparable ($P_{max} = (1.5 \pm 1.0) \times 10^{-14}$ J s$^{-1}$ in without crosslinkers; $P_{max} = (1.2 \pm 1.0) \times 10^{-14}$ J s$^{-1}$ in fascin) (Fig. 2f, g). In contrast, mixed polar bundles with fimbrin perform significantly lower power ($P_{max} = (1.2 \pm 0.9) \times 10^{-15}$ J s$^{-1}$ in fimbrin).

Furthermore, the ATP consumption rate was differentially controlled depending on the polarity of the bundles. The contractile unipolar bundles consumed ATP as rapidly as non-crosslinked network ($\Delta[ATP]/\Delta t = 50.2 \pm 12.0$ μM min$^{-1}$ in without crosslinkers; $\Delta[ATP]/\Delta t = 41.9 \pm 14.5$ μM min$^{-1}$ in fascin) (Fig. 2h, i). In contrast, the ATP consumption rate of the mixed polar bundles was significantly slower than that of the unipolar bundles and the non-crosslinked network ($\Delta[ATP]/\Delta t = 18.5 \pm 5.1$ μM min$^{-1}$ in fimbrin) (Fig. 2h, i). This result suggests that the myosin thick filament is trapped within the mixed polarity bundles[16], applying significant tension within the network and resulting in slower ATP hydrolysis due to its load dependence[48,49]. We confirmed that α-actinin-crosslinked networks (0.7 μM α-actinin; [Actin]:[α-actinin] = 1:0.12) can contract and consume ATP as quickly as the non-crosslinked network ($P_{max} = (1.7 \pm 1.1) \times 10^{-14}$ J s$^{-1}$ and $\Delta[ATP]/\Delta t = 46.0 \pm 18.1$ μM min$^{-1}$ in α-actinin) (Fig. 2f–i). One possibility to explain this phenomenon is that the narrow F-actin spacing (fimbrin ~6 nm; α-actinin ~35 nm[59]) induces the load-dependent behavior of myosin[16]. In mixed polar bundles, the efficiency of the energy conversion is lower than the other crosslinked networks, attributed to significantly lower power (Supplementary Fig. 8). We confirmed that the ATP consumption rate of crosslinked actin networks without myosin present was comparable to the ATP consumption rate of non-crosslinked actin networks; thus, the variation in ATP consumption rates of actomyosin networks with different crosslinkers is predominantly attributable to myosin-based ATP consumption (Supplementary Fig. 9, Movie S7). Together, these results demonstrate that polarity of bundling can significantly modulate both ATP consumption and mechanical work, differentially regulating the efficiency of contraction.

## Branched networks preferentially dissipate energy and linear networks perform work

We have demonstrated that polarity of F-actin organized by crosslinkers can modulate contractile behaviors of the actomyosin network as well as energy consumption rates. On the other hand, F-actin architecture can be controlled through F-actin nucleators, such as a branched F-actin network nucleator Arp2/3 complex[60] and a linear F-actin nucleator formin mDia1[61] (Fig. 3a). Notably, the Arp2/3-nucleated network (300 nM Arp2/3) prevents network contraction and significantly reduces mechanical power ($P_{max} = (1.5 \pm 1.0) \times 10^{-14}$ J s$^{-1}$ in spontaneously polymerized network without nucleators; $P_{max} = (4.0 \pm 1.8) \times 10^{-15}$ J s$^{-1}$ in Arp2/3) (Fig. 3b, e–g, Movie S8 and S9), yet the ATP consumption rate remains comparable to that of the spontaneously polymerized network ($\Delta[ATP]/\Delta t = 47.4 \pm 10.8$ μM min$^{-1}$ in without nucleators; $\Delta[ATP]/\Delta t = 44.9 \pm 11.4$ μM min$^{-1}$ in Arp2/3) (Fig. 3h, i). The Arp2/3 network exhibited spatially nonuniform compressive and extensile strain due to the Arp2/3-nucleated 'flower-like' networks diffuse in space without global contraction (Fig. 3e). These results

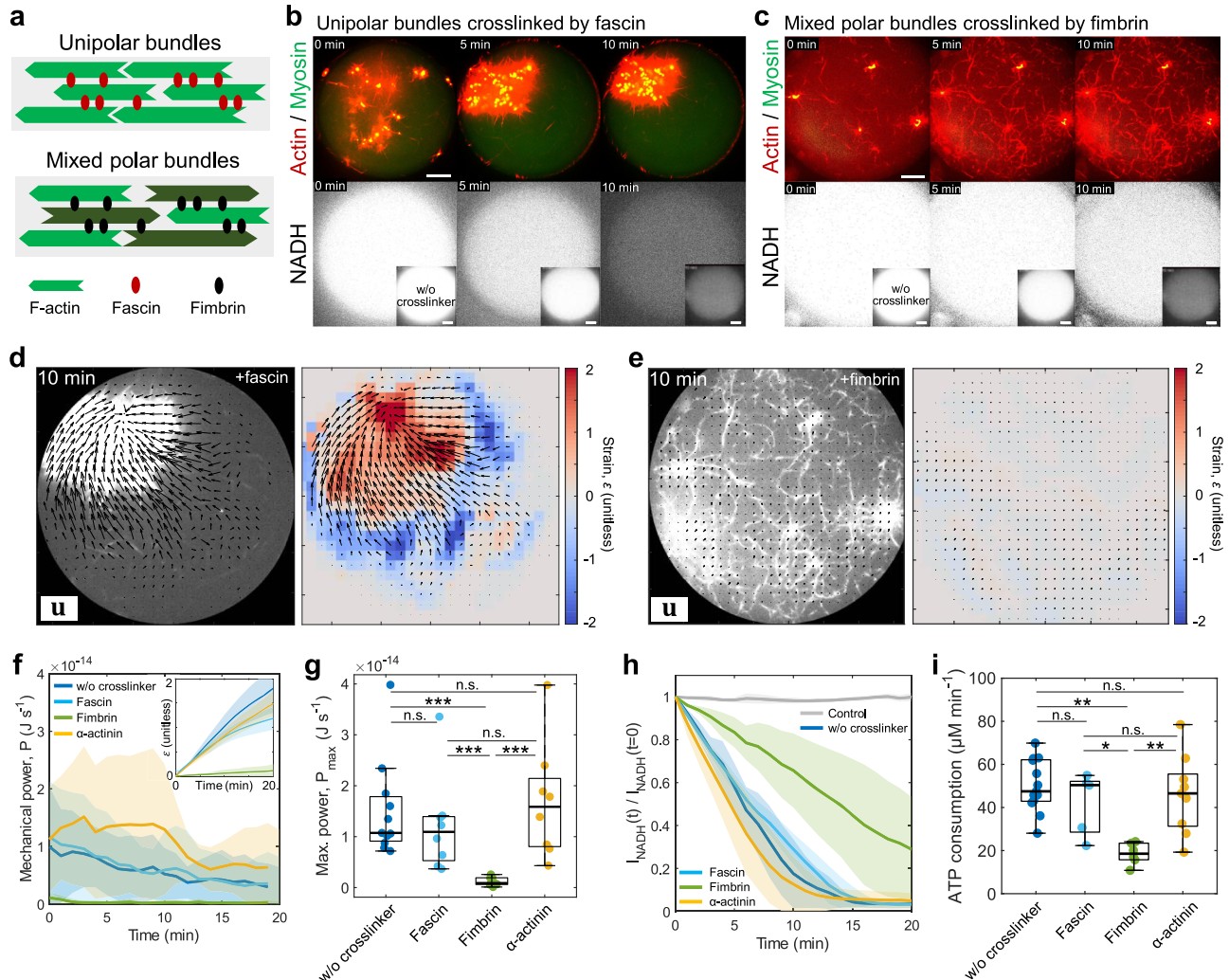

**Fig. 2 | Polarity of F-actin bundles controls mechanical power and ATP consumption rate. a** Schematic showing the unipolar bundles crosslinked by fascin (top) and mixed polar bundles crosslinked by fimbrin (bottom). **b, c** Time-lapse images showing the contraction of the actin network crosslinked by fascin (**b**, 1 μM) and fimbrin (**c**, 1 μM) (actin in red, myosin in green), and the NADH fluorescence (in white). **d, e** PIV analysis on actomyosin network contraction for fascin (**d**) and fimbrin-crosslinked network (**e**). Black vectors are the total displacement over 10 min and vector magnitudes are normalized by the maximum displacement (left). The colormap represents local strain fields (right). **f** Instantaneous power over time ($n = 12$ droplets and $N = 10$ independent experiments in without crosslinkers; $n = 8$ and $N = 7$ in fascin; $n = 6$ and $N = 6$ in fimbrin; $n = 6$ and $N = 3$ in α-actinin). Inset shows mean compressive strain over time. **g** Boxplot showing the maximum instantaneous power performed by myosin

extracted from (**f**) ($n = 12$ droplets and $N = 10$ independent experiments in without crosslinkers; $n = 8$ and $N = 7$ in fascin; $n = 7$ and $N = 7$ in fimbrin; $n = 8$ and $N = 4$ in α-actinin). **h** NADH fluorescence over time normalized by the initial time point ($n = 7$ and $N = 4$ in control; $n = 9$ and $N = 3$ in without crosslinkers; $n = 5$ and $N = 3$ in fascin; $n = 6$ and $N = 3$ in fimbrin; $n = 7$ and $N = 4$ in α-actinin). **i** Boxplot showing the ATP consumption rate in (**h**) ($n = 13$ and $N = 4$ in without crosslinkers; $n = 5$ and $N = 3$ in fascin; $n = 6$ and $N = 3$ in fimbrin; $n = 9$ and $N = 4$ in α-actinin). Scale bars, 10 μm. Data are presented as mean ± SD (**f, h**) or boxplots where the interquartile range (IQR) is between Q1 (25th percentile) and Q3 (75th percentile), the center line indicates the median, whiskers are extended to Q3 + 1.5 × IQR and Q1 − 1.5 × IQR (**g** and **i**); **p < 0.01; ***p < 0.001 in a two-sided Wilcoxon rank sum test. n.s., not significant. Source data are provided as a Source Data file.

indicate that myosin ATP hydrolysis are not hindered by the branched network; instead, the dense branched network may dissipate energy by reducing the force per filament applied by myosin, or possibly due to debranching/severing, as suggested by numerical simulations[45,55,56].

We confirmed that even at 60 nM Arp2/3 ([Arp2/3]/[Actin] = 0.01), network contraction was effectively prevented while maintaining an ATP consumption rate as fast as that without Arp2/3 (Supplementary Fig. 10, Movie S10). By contrast, network contraction was observed at 6 nM Arp2/3 ([Arp2/3]/[Actin]=0.001), where mechanical power was lower than that without Arp2/3, along with a slowed ATP consumption rate compared to 60 nM Arp2/3 and the condition without Arp2/3 (Supplementary Fig. 10, Movie S10). This result indicates that at a lower concentration of Arp2/3, Arp2/3 branching partially functions as a crosslinker, inducing load-dependent ATP hydrolysis of myosin within

contracted actin aggregates. Together, these results suggest that the degree of branching via Arp2/3 is crucial in regulating contractile behavior and myosin ATP consumption.

In contrast, mDia1-nucleated networks (300 nM mDia1) form small clusters that eventually merge, corresponding to a local compressive strain field, suggesting efficient contraction and mechanical power higher than the Arp2/3 network ($P_{max} = (7.0 \pm 3.6) \times 10^{-15}$ J s$^{-1}$ in mDia1) (Fig. 3c, e–g, Movie S8 and S9). However, the ATP consumption rate of the mDia1-nucleated network dramatically slowed down ($\Delta[ATP]/\Delta t = 23.2 \pm 6.8$ μM min$^{-1}$ in mDia1) (Fig. 3h, i). This result may suggest that the load-dependence of myosin ATP hydrolysis comes into play within the small contracting asters[48,49]. We confirmed that the ATP consumption rate of actin in the presence of Arp2/3 and mDia1 was comparable to the ATP consumption rate of a spontaneously

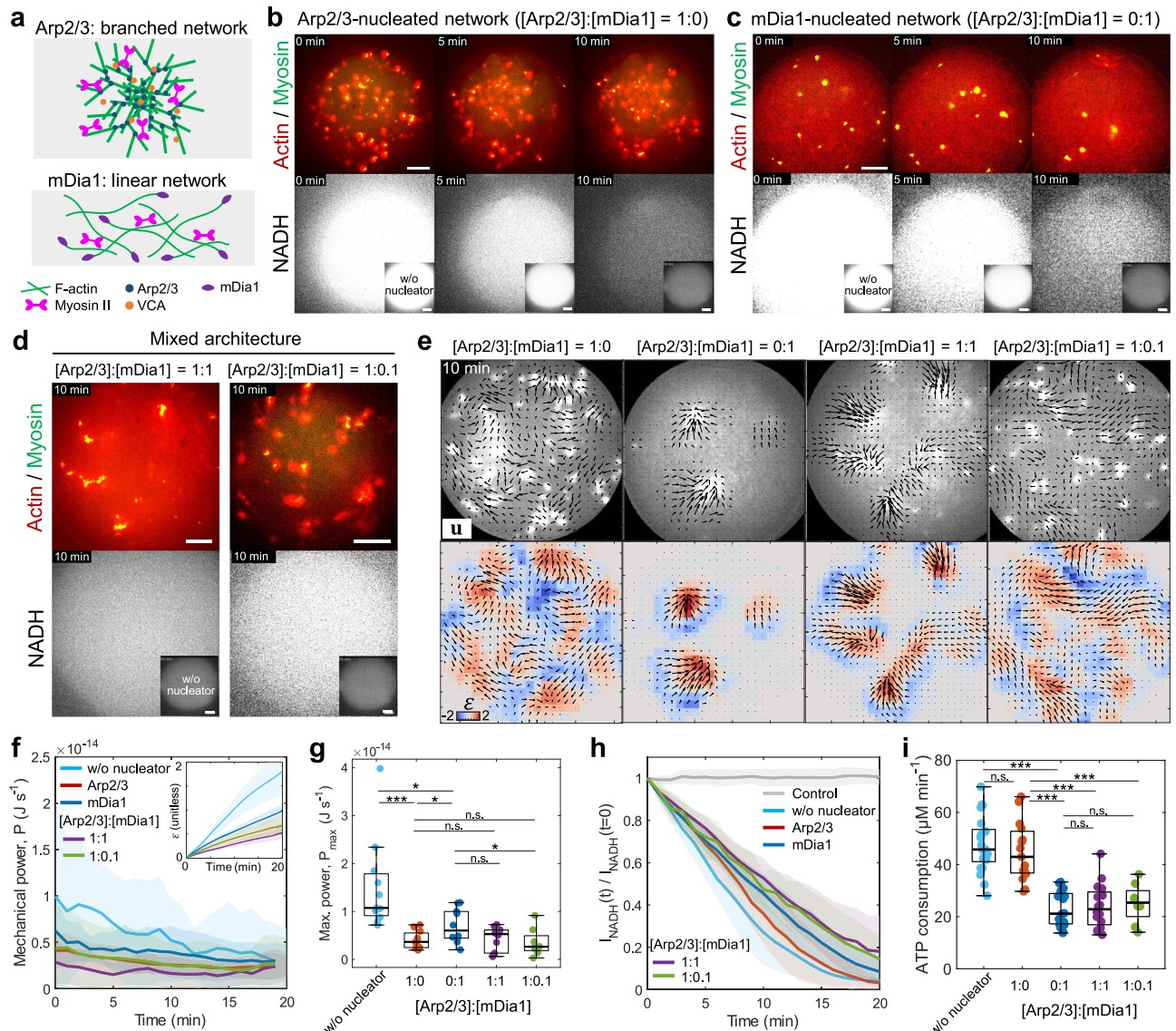

**Fig. 3 | Branched and linear F-actin architecture alter mechanical power and ATP consumption rate. a** Schematic showing the Arp2/3-nucleated branched network (top) and the mDia1-nucleated linear network (bottom). **b, c** Time-lapse images showing the contraction of the actin network nucleated by the Arp2/3 complex (**b**, 300 nM Arp2/3) and mDia1 (**c**, 300 nM mDia1) (actin in red, myosin in green), and the NADH fluorescence (in white). **d** Snapshots showing the contraction of the mixed architecture nucleated by [Arp2/3]:[mDia1] = 1:1 (left) and [Arp2/3]:[mDia1] = 1:0.1 (right) and the NADH fluorescence (bottom) at 10 min. The total nucleator concentration is fixed at 300 nM. **e** Actomyosin network contraction (top) for different Arp2/3 to mDia1 ratio. Black vectors are the total displacement over 10 min and vector magnitudes are normalized by the maximum displacement. The colormap represents local strain fields (bottom). **f** Instantaneous power over time ($n = 12$ droplets and $N = 11$ independent experiments in without nucleators; $n = 10$ and $N = 9$ in Arp2/3 only; $n = 9$ and $N = 7$ in mDia1 only; $n = 9$ and $N = 6$ in [Arp2/3]:[mDia1] = 1:1; $n = 8$ and $N = 4$ in [Arp2/3]:[mDia1] = 1:0.1). Inset shows the mean compressive strain over time. **g** Boxplot showing the maximum

instantaneous power performed by myosin extracted from (f) ($n = 11$ droplets and $N = 10$ independent experiments in without nucleators; $n = 11$ and $N = 10$ in Arp2/3 only; $n = 10$ and $N = 8$ in mDia1 only; $n = 10$ and $N = 7$ in [Arp2/3]:[mDia1] = 1:1; $n = 8$ and $N = 4$ in [Arp2/3]:[mDia1] = 1:0.1). **h** NADH fluorescence over time normalized by the initial time point ($n = 12$ and $N = 6$ in control; $n = 17$ and $N = 7$ in without nucleators; $n = 15$ and $N = 9$ for Arp2/3 only; $n = 20$ and $N = 10$ in mDia1 only; $n = 16$ and $N = 8$ in [Arp2/3]:[mDia1] = 1:1; $n = 8$ and $N = 4$ in [Arp2/3]:[mDia1] = 1:0.1). **i** Boxplot showing the ATP consumption rate in (**h**) ($n = 12$ and $N = 6$ in control; $n = 21$ and $N = 9$ in without nucleators; $n = 15$ and $N = 9$ for Arp2/3 only; $n = 20$ and $N = 10$ in mDia1 only; $n = 16$ and $N = 8$ in [Arp2/3]:[mDia1] = 1:1; $n = 8$ and $N = 4$ in [Arp2/3]:[mDia1] = 1:0.1). Scale bars, 10 μm. Data are presented as mean ± SD (**f, h**) or boxplots where the interquartile range (IQR) is between Q1 (25th percentile) and Q3 (75th percentile), the center line indicates the median, whiskers are extended to Q3 + 1.5 × IQR and Q1 − 1.5 × IQR (**g, i**); $*p < 0.05$; $***p < 0.001$ in a two-sided Wilcoxon rank sum test. n.s., not significant. Source data are provided as a Source Data file.

polymerized network ($\Delta[\text{ATP}]/\Delta t = 4.3 \pm 1.6\ \mu\text{M min}^{-1}$) (Supplementary Fig. 3, Movie S11).

Interestingly, mixed architectures containing both Arp2/3 and mDia1 exhibit mechanical power comparable to the purely branched network (the total nucleator concentration is fixed at 300 nM; $P_{max} = (4.3 \pm 2.5) \times 10^{-15}\ \text{J s}^{-1}$ at [Arp2/3]:[mDia1] = 1:1, $P_{max} = (3.5 \pm 2.8) \times 10^{-15}\ \text{J s}^{-1}$ at [Arp2/3]:[mDia1] = 1:0.1) (Fig. 3d–g, Movie S8 and S9). In contrast, the ATP consumption rates significantly

slow down compared to the purely branched network for both 1:1 and 1:0.1 Arp2/3 to mDia1 concentration ratios ($\Delta[\text{ATP}]/\Delta t = 24.0 \pm 8.5\ \mu\text{M min}^{-1}$ at [Arp2/3]:[mDia1] = 1:1; $\Delta[\text{ATP}]/\Delta t = 25.1 \pm 7.5\ \mu\text{M min}^{-1}$ at [Arp2/3]:[mDia1] = 1:0.1) (Fig. 3h, i). The Arp2/3-nucleated branched network exhibits low energy conversion efficiency, while the linear network nucleated by mDia1 enhances efficiency when mixed with the branched network (Supplementary Fig. 8). These results suggest that mDia1-nucleated linear networks may dominate load-dependent ATP

hydrolysis even at a low concentration, whereas mechanical power could be attenuated by branched networks nucleated by Arp2/3. Together, these results indicate that branched Arp2/3 networks work as medium to dissipate energy, while linear mDia1 networks form a contractile network with slower energy consumption rate.

### The dynamic steady-state maintained by F-actin severing regulates ATP consumption rate

We have demonstrated that structural aspects, such as F-actin architecture and bundling, can regulate myosin ATP consumption and mechanical work. In cells, these structures undergo disassembly and transform into different F-actin architectures. Therefore, our interest lies in understanding the energetic cost of maintaining a dynamic steady state without sustaining any specific structure. It is important to clarify that when we refer to the 'dynamic steady-state' in our study, we specifically focus on the propensity to retain a constant F-actin density and not condense (contract). In our context, the term 'dynamic steady-state' describes a state in which there are no contracted actomyosin aggregates present. This is crucial because the presence of contracted actomyosin increases local actin density, motion of myosin is confined, potentially influencing the load-dependent ATP hydrolysis rate of myosin.

To clarify this point, we define the 'dynamic steady-state' as a state that satisfies the following conditions: (i) the macroscopic strain rate of the actin network is zero ($\dot{\varepsilon} \equiv -\langle \nabla \cdot \mathbf{v} \rangle = 0$) and (ii) the spatially averaged magnitude of velocity takes a finite value ($\langle |\mathbf{v}| \rangle > 0$)[9,62,63]. We demonstrated that the gelsolin-severed system exhibits a strain rate fluctuating around zero with a finite magnitude of velocity (Supplementary Fig. 11). Therefore, the gelsolin-severed actomyosin system can be characterized as being in a dynamic steady state. By contrast, the cofilin-severed network exhibits a finite total strain due to the initial contraction and the formation of actin aggregates, suggesting that cofilin-mediated severing is insufficient to achieve a dynamic steady state under the present experimental conditions. Additionally, when the network is stabilized by phalloidin, the strain rate takes a large value due to the presence of macroscopic contractile flow (Supplementary Fig. 11). Consequently, the phalloidin-stabilized actomyosin system does not satisfy the dynamic steady-state condition. It is worth noting that the molecular origin of the dynamic steady-state lies in the nature of skeletal muscle myosin II, which is known for its fast motor activity and low duty ratio (~0.05[38]), resulting in rapid binding and unbinding cycles to F-actin. This rapid cycling induces significant fluctuations of F-actin perpendicular to the filament axis without causing macroscopic contraction, named as 'plucking'[62]. These reversible F-actin plucking events arise from transient and diverse interactions between non-aligned myosin and F-actin, and they occur regardless of myosin isoforms[62]. The plucking mode of myosin contraction has been theoretically studied and well-established[41,64].

To investigate how much ATP is consumed in a dynamic steady-state, we employed the actin-severing proteins gelsolin and cofilin. A single gelsolin molecule severs F-actin upon binding to it and caps it[65]. On the other hand, multiple cofilin molecules bind to F-actin, resulting in severing at the domain boundary of the cofilin-coated region[66]. We aim to understand how these different molecular mechanisms of severing influence myosin ATP consumption rate. We chose a cofilin concentration comparable to that of actin, as demonstrated by the previous study that the influence of cofilin on actin turnover saturates at [Actin]:[Cofilin]=1:1[67]. On the other hand, even at a low concentration of gelsolin (56 nM gelsolin; [Actin]:[Gelsolin]=1:0.01), it severs the filaments, allowing myosin to freely diffuse over space without its motion being constrained by actin aggregates (Supplementary Fig. 11, Movie S12). As we can prevent the formation of actin aggregates and achieve a dynamic steady-state for myosin motion at a low concentration of gelsolin, we have opted to maintain this gelsolin concentration for our investigation.

The addition of gelsolin (56 nM gelsolin; [Actin]:[Gelsolin] = 1:0.01) or cofilin (6 μM cofilin; [Actin]:[Cofilin] = 1:1) both prevented the formation of large actin aggregates, where multiple small aggregates randomly distributed in space (Supplementary Fig. 12, Movie S12). However, gelsolin-mediated severing led to the drastic depolymerization of the network, allowing myosin-thick filaments to freely diffuse. In contrast, cofilin-mediated severing preserved the F-actin network, trapping myosin thick filaments within it (Supplementary Fig. 12, Movie S12). Notably, the gelsolin-severed network consumed more ATP than the contractile non-severed network ($\Delta[\text{ATP}]/\Delta t = 33.4 \pm 9.6\ \mu\text{M min}^{-1}$ in non-severed with 50 nM myosin; $\Delta[\text{ATP}]/\Delta t = 40.3 \pm 10.2\ \mu\text{M min}^{-1}$ in gelsolin with 50 nM myosin), indicating that the dynamic steady-state may counteract the effect of load-dependent ATP hydrolysis (Supplementary Fig. 12).

In contrast, the cofilin-severed network consumed ATP slower than the non-severed network ($\Delta[\text{ATP}]/\Delta t = 19.1 \pm 7.1\ \mu\text{M min}^{-1}$ at [Actin]:[Cofilin] = 1:1) (Supplementary Fig. 12). We suspect that the binding of cofilin to F-actin may compete with myosin binding and reduces the apparent ATPase activity, as cofilin is known to alter the F-actin pitch and competes with the binding of other actin-binding proteins[68–70]. Alternatively, the dynamic steady-state without forming actin aggregates could be achieved at high KCl concentration, as high salt condition prevents myosin thick filament formation[71]. However, we confirmed that the high salt condition also decelerates ATP consumption rates of both the full-length myosin and heavy meromyosin (in full-length myosin at 0.2 μM, $\Delta[\text{ATP}]/\Delta t = 101.0 \pm 30.8\ \mu\text{M min}^{-1}$ at 46 mM KCl and $[\text{ATP}]/\Delta t = 5.9 \pm 1.9\ \mu\text{M min}^{-1}$ at 460 mM KCl; in heavy meromyosin at 0.2 μM, $\Delta[\text{ATP}]/\Delta t = 34.0 \pm 11.6\ \mu\text{M min}^{-1}$ at 46 mM KCl and $[\text{ATP}]/\Delta t = 10.1 \pm 3.1\ \mu\text{M min}^{-1}$ at 460 mM KCl) (Supplementary Fig. 13); thus, the examination of dynamic steady-state by high salt condition was prohibited as we could not independently change the myosin thick filament formation and ATP consumption rates. Together, our results suggest that gelsolin is a better-suited severing protein for maintaining a dynamic steady-state.

Conversely, F-actin stabilization by phalloidin significantly decelerated the ATP consumption rates of myosin than that of the non-stabilized network ($\Delta[\text{ATP}]/\Delta t = 43.0 \pm 4.7\ \mu\text{M min}^{-1}$ in non-stabilized network with 50 nM myosin; $\Delta[\text{ATP}]/\Delta t = 17.0 \pm 8.3\ \mu\text{M min}^{-1}$ at 50 nM myosin with 17 μM phalloidin) (Supplementary Fig. 14, Movie S13). Since phalloidin inhibits depolymerization and stiffens F-actin[72], this result suggests that F-actin turnover may relax the stress accumulated within the aggregates via depolymerization and filament breakage[45,63], by which the load-dependence of myosin could be released in a non-stabilized network. Together, these findings suggest that the dynamic steady-state maintained by F-actin severing can enhance myosin energy consumption rate via releasing the load-dependence of myosin, which is energetically more costly than maintaining contractile F-actin structures[62].

## Discussion

In this study, we developed an experimental platform to simultaneously quantify F-actin architectures through live imaging and ATP consumption rates using the NADH-coupled assay. With this system, we investigated the influence of F-actin architecture, bundling, and severing-induced dynamic steady-state on the energetic efficiency of myosin force generation. Myosin ATP consumption rate was differentially regulated by distinct F-actin architectures, such as unipolar and mixed polar networks, as well as branched or linear actin network (Fig. 4a). Furthermore, maintaining the dynamic steady-state of myosin motion without confining myosin within actin aggregates consumed a larger amount of ATP. Together, we demonstrate that the mechanical work and ATP consumption rate of myosin can be tuned several-fold, solely based on F-actin architecture at a constant concentration of actin and myosin (Fig. 4b, c). Thus, this study establishes a link between F-actin architecture and myosin II-based energy

**Fig. 4 | F-actin architecture controls the ATP consumption rate of myosin II. a** The summary of the architectural regulation of ATP consumption. The mechanical work and ATP consumption rate of myosin II in crosslinked bundles are controlled by the polarity of the bundle structure. Unipolar bundles enable contraction while consuming a large amount of ATP, whereas mixed polarity bundles suppress both mechanical work and ATP consumption rates. On the other hand, the branched network dissipates mechanical work while allowing for high ATP consumption. In contrast, a linear actin network allows for contraction while slowing down ATP consumption. The dynamic steady-state maintained by gelsolin-based severing enhances ATP consumption rates. **b** Maximum instantaneous power dependence on ATP consumption rates ($n = 11$ droplets and $N = 10$ independent experiments in control (spontaneously polymerized non-crosslinked network at 50 nM myosin); $n = 6$ and $N = 4$ in control at 12.5 nM myosin; $n = 6$ and $N = 3$ in fascin; $n = 5$ and $N = 3$ in fimbrin; $n = 6$ and $N = 3$ in $\alpha$-actinin; $n = 6$ and $N = 3$ in Arp2/3; $n = 6$ and $N = 4$ in mDia1; $n = 9$ and $N = 6$ in [Arp2/3]:[mDia1]=1:1 mixed network). The dashed line represents the fitting between control (filled circle) and control at 12.5 nM myosin (empty circle). The large points are mean ± SD. Individual data is shown as small points. **c** Schematic summarizing the F-actin architectural control of mechanical power and ATP consumption rate by myosin. Source data are provided as a Source Data file.

consumption, underscoring the importance of regulating myosin energy consumption through distinct F-actin architectures.

Recent studies have demonstrated the significant role of bundled F-actin networks in regulating myosin thick filament motion in a 2D open system[16]. Fascin-bundled unipolar networks enable persistent unidirectional motion of myosin thick filaments, while fimbrin-bundled mixed polar networks restrict their motion. Our present study reveals that unipolar bundled networks behave similarly to non-crosslinked networks in terms of both high mechanical power and rapid ATP consumption. In contrast, mixed polar bundled networks not only hinder network contraction but also significantly reduce myosin ATP consumption, implying a load-dependent ATP hydrolysis mechanism for myosin[48,49]. In this scenario, myosin-induced tension is stored within the mixed polar bundled networks, allowing for high tension and reduced ATP consumption. This architectural energy regulation may be preferred for stress fibers, where F-actin polarity alternates with submicron periodicity in stably adherent non-migrating cells[73].

It is noteworthy that mixed polar architecture in non-crosslinked bundles has been shown to facilitate network contraction[13]. In contrast, in our study, fimbrin-crosslinked mixed polar bundles impede network contraction, highlighting the significance of actin crosslinking in controlling mechanical performance. On the other hand, $\alpha$-actinin-crosslinked mixed polar bundles allowed significantly higher mechanical work and ATP consumption rate than that of fimbrin-crosslinked bundles, indicating that the F-actin spacing plays key roles in energy conversion of myosin motors[16], while the contractility may depend on myosin concentration and actinin/actin ratio[8]. Additionally, previous research indicates that fascin-bundled unipolar networks exhibit greater contractility compared to cortexillin-I-bundled mixed polar networks at the same crosslinker concentration, while contractile behaviors may depend on crosslinker concentration[14]. It should be noted that we chose sufficiently high concentrations of fascin, fimbrin, and $\alpha$-actinin (5–10 mol% relative to actin) to impact myosin force generation, as demonstrated in previous studies[16].

On the other hand, it has been demonstrated that branched and linear F-actin networks differentially regulate the contractile behaviors of myosin thick filaments in a 2D open system[11]. Spontaneously polymerized networks and linear networks nucleated by mDia1 promote myosin contraction, whereas branched networks nucleated by the Arp2/3 complex hinder contraction due to branching-induced attenuation of the contractile force of myosin. Our study reveals that branching not only suppresses myosin motion but also leads to high ATP consumption. This suggests that the power stroke of myosin is not inhibited within the branched network; instead, the energy is dissipated through the branches, possibly due to debranching/severing, as suggested by numerical simulations[45,55,56]. Given that the cell cortex is primarily composed of Arp2/3-nucleated filaments (~80%[74]), it may imply that such a structure can effectively attenuate the mechanical work of myosin, thereby preventing undesired contractile deformation.

In contrast, we have demonstrated that the ATP consumption rate of linear networks can be significantly lower than that of spontaneously polymerized networks, despite both architectures allowing contraction, as observed in previous studies[11]. Thus, while the spontaneously polymerized network and the linear network nucleated by mDia1 share similar F-actin architectures, our results suggest that nucleators can play a significant role in controlling myosin ATP consumption rates. We suspect that the reduced ATP consumption in the mDia1 network may result from the load-dependence of myosin ATP hydrolysis, where higher loads decelerate ATP consumption rates and generate greater forces[48,49]. Previous research has shown that myosin contraction and formin-based F-actin nucleation can cooperatively generate contracting polar-aster-like structures in cells[75]. Our results imply that such aster structures could be highly efficient with both increasing contractile efficiency and reducing myosin ATP consumption rates.

Interestingly, the ATP consumption rate of the mixed architecture formed by the branched and linear network was significantly decelerated, even at 1:0.1 Arp2/3 to mDia1 ratio. This result suggests that the influence of the long linear mDia1-nucleated filaments propagates over long distance within the network. Cooperative F-actin assembly of Arp2/3 and mDia1 have been reported in the cell cortex[76]. It is of note that the cell cortex is primarily composed of Arp2/3-nucleated branched filaments, while mDia1-nucleated linear filaments make up only 10% of the total number of filaments[74]. Nevertheless, mDia1-nucleated filaments have been shown to be a determinant of the cortical tension and elasticity[74,77]. Thus, our results may suggest that not only the mechanical properties of the cortex but also energy consumption could be controlled by linear actin networks in the cell cortex.

In cells, distinct F-actin architectures are chosen to fulfill specific biological functions, such as branched F-actin networks in retrograde actin flow during cell migration[20], formin-nucleated contractile rings during cell division[78], and protruding F-actin structures like fascin-crosslinked filopodia[79] or fimbrin-crosslinked microvilli[80]. The present study suggests that these distinct F-actin architectures could differentially regulate the ATP consumption rates of myosin II, highlighting the importance of the energetic relationship between F-actin architecture and myosin II energy consumption. For instance, we have demonstrated that both branched and mixed polarity bundled networks significantly reduce mechanical work. However, while the branched network consumes a large amount of energy, the mixed polarity network can reduce energy consumption. Nevertheless, cells often choose the energetically more costly branched network nucleated by Arp2/3, as it could be more easily reorganized and reassembled compared to tightly crosslinked bundles. Such differential regulation of mechanical work and energy consumption through nucleators and crosslinkers may be chosen by cells to achieve specific cellular functions. Thus, our study has implications for how distinct F-actin architectures are selected to achieve desired cellular mechanical functions,

potentially providing a way to control transitions between different cellular states based on the energetic principles of F-actin structure formation.

It is of note that F-actin architecture continually transitions within the cell, as observed in structures like filopodia protruding from lamellipodia. For filopodia formation, the 'convergent elongation model' has been proposed, wherein the branched F-actin network acts as a scaffold for filopodial protrusion from lamellipodia[81]. Filopodia consists of parallel actin bundles crosslinked by fascin[79]. Thus, myosin motors in lamellipodia may be exposed to a structural transition from a branching to a bundling architecture. Our study demonstrated that the Arp2/3 network impedes myosin contraction, while fascin-crosslinked parallel bundles facilitate myosin contraction. Consequently, while myosin motors in lamellipodia contracts moderately, upon reaching filopodia, they may induce rapid contraction of filopodia. Thus, the structural shift from branched to bundled networks could be crucial in the sequential processes observed in adherent migratory cells: extending the leading edge, forming substrate adhesion, and rapidly contracting filopodia[82]. The variation in ATP consumption rates at different cellular locations might play a role in guiding structure formation.

It should be noted that the range of ATP consumption rate in our reconstituted system is comparable to cellular ATP consumption rate. For instance, considering the minimum estimate of the total number of 43,800 non-muscle myosin IIA (NMIIA) bipolar filaments contained in the retrograde flow of the lamellipodia of U2O2 cells[83] and ATPase activity of 0.49 ATP s$^{-1}$ per NMIIA monomer[84], we estimate the myosin IIA ATP consumption rate per single cell to be $\sim 4.3 \times 10^4$ ATP s$^{-1}$ cell$^{-1}$. Given that the typical U2O2 cell volume is ~ 4000 μm$^3$[85,86], this lower bound estimate is comparable to the ATP consumption rate of ~1 μM min$^{-1}$ in a droplet with a diameter of 20 μm in this study. Thus, we consider that the energetic relationship between F-actin architecture and myosin II energy consumption rates established in this study could be applicable to cellular situations.

Also, it is worth noting that the total ATP consumption rate within a cell is primarily dominated by metabolism, including protein synthesis. Previous estimates suggested that the total ATP consumption rate due to cellular metabolism would be $\sim 10^7 - 10^8$ ATP s$^{-1}$ cell$^{-1}$[87,88], which is more than three orders of magnitude larger than the ATP consumption rate of myosin II in cells. Nevertheless, the difference in ATP consumption rates could potentially serve as a reference in cells for selecting which F-actin architectures are more efficient for achieving specific mechanical functions. To further understand how cells utilize F-actin architectural regulation of myosin ATP consumption, investigating the energetic relationship by isolating ATP consumption related to mechanics from that associated with metabolism within cells will be an important future challenge.

In physiological conditions, myosin-induced mechanical power could be smaller than our estimates. Given that the macromolecular crowding may further increase the viscosity of the cytoplasm, the contraction of the actomyosin network could be slowed down by the larger fluid drag[89]. Furthermore, macromolecular crowding also increases the local protein concentration, potentially influencing the F-actin architecture through bundling[90,91]. Additionally, protein-protein interactions apply a friction force, counteracting the direction of the actomyosin contraction[1,92]. It will be an interesting avenue for future work to explore the impact of such physiological conditions by varying the viscosity of the solution or introducing actin-membrane binding to induce protein-protein interaction-based friction[93]. Moreover, microtubule networks in cells can also interfere the contraction of the actomyosin networks[94], where investigation of the microtubule-actin composite networks in vitro will provide valuable insights[15]. The present system serves as a versatile platform for quantitatively studying these effects on mechanical work and ATP consumption rate.

In conclusion, we have demonstrated that F-actin architectural control over branching or polarity of the bundles differentially regulates ATP consumption rates, while maintaining the dynamic steady-state of myosin motion through actin severing consumes more energy than sustaining the structures. The regulation of myosin II ATP consumption rates by F-actin architecture has implications for regulating cellular structures based on energetics, highlighting the potential importance of the coupling between F-actin architecture and myosin II energy consumption. Furthermore, these energetic principles could potentially be applicable for developing design principles of protein-based soft robotics, such as actomyosin-based actuators[95,96]. The F-actin architecture could be optimized to achieve the highest energy conversion efficiency of microrobots by measuring their ATP consumption and work production. Thus, this study provides a simple yet versatile platform for investigating the quantitative relationship between F-actin architecture and myosin II-based ATP consumption rates in various systems in the fields of cell biology and biophysics, as well as in materials science.

## Methods

### Lipid compositions
Lipids are a combination of L-α-phosphatidylcholine from egg yolk at 61% (EPC) (840051 C; Avanti), cholesterol at 36% (ovine wool) (110796; Avanti), and 1,2-distearoyl-sn-glycero-3-phosphoethanolamine-N-[methoxy(polyethylene glycol)−2000] at 2% (18:0 PEG2000 PE) (880120; Avanti) was used to prevent non-specific adsorption of proteins at the water-oil interface, Oregon Green 1,2-Dihexadecanoyl-snGlycero-3-Phosphoethanolamine (DHPE) (O12650; Invitrogen) at 1%. Lipids were dissolved in chloroform solution and stored at −20 °C. The lipids were combined in a glass vial and dried under Ar gas. The chloroform is dried, and the lipids are dissolved in mineral oil (Sigma-Aldrich) at 2 mg ml$^{-1}$. The oil mixture is then sonicated in a bath sonicator for 2 h at room temperature. The mixture is then stored at 4 °C up to a week.

### Buffers
Buffer concentrations were referenced from previously developed actomyosin assays[24,97]. Actin polymerization buffer contains 2 mM CaCl2, 6 mM MgCl2, 50 mM KCl, 10 mM HEPES (pH 7.6), 0.8 mM DTT, 5 mM ATP, 50 mg ml$^{-1}$ dextran, and 175 mM sucrose. The inclusion of dextran and sucrose is a result of previously optimized conditions in liposome systems[24,97]. The storage buffer for actin (G-buffer) contains 2 mM Tris-HCl (pH 8.0) and 0.1 mM CaCl2, 0.2 mM ATP, and 0.5 mM DTT, and 22 μM actin. The storage buffer for myosin contains 0.5 M KCl, 0.1 M HEPES (pH 7.0), and 0.76 μM myosin. The storage buffer for NADH contains 100 mM NaHCO3 and 14 mM NADH. The storage buffer for PEP contains 100 mM NaHCO3 and 48.5 mM PEP. The storage buffer for PK contains 10 mM HEPES (pH 7.4), 100 mM KCl, and 2 KU mL$^{-1}$ PK. The storage buffer for LDH contains 10 mM HEPES (pH 7.4), 100 mM KCl, and 2.6 KU mL$^{-1}$ LDH. All the regents were purchased from Sigma-Aldrich.

### NADH-coupled assay compositions
Nicotinamide adenine dinucleotide (NADH)-coupled assay consists of pyruvate kinase (PK) and phosphoenolpyruvate (PEP)-based ATP regeneration system, together with lactic dehydrogenase (LDH) that oxidize NADH[27,28] (Fig. 1b). The final concentration of NADH (10128023001; Roche), PEP (10108294001; Roche), PK (P9136; Sigma-Aldrich) and LDH (L1254; Sigma-Aldrich) is 0.5 mM, 1 mM, 10 U mL$^{-1}$, and 16 U mL$^{-1}$, respectively. The choice of NADH concentration is based on technical considerations. First, we selected a sufficiently low NADH concentration to guarantee the complete decay of NADH fluorescence in ~20 min at [Myosin] = 50 nM. This enhances the visibility and clarity in identifying ATP consumption. More importantly, as the conversion of NADH requires a comparable amount of PEP and the associated

concentrations of PK and LDH, using a larger NADH concentration would significantly increase the overall concentration of the additional enzymatic system. To minimize the impact of NADH assay-related enzymatic systems on the conventional actin polymerization buffer, we selected a low enough NADH concentration for our measurements.

### Protein concentrations
The final concentration of non-fluorescent actin (AKL99-D; Cytoskeleton Inc.) rhodamine-actin (AR05-C; Cytoskeleton Inc.) and skeletal muscle myosin II (MY02; Cytoskeleton Inc.) were 5.3 μM, 0.7 μM, and 12.5–50 nM, respectively. In the experiments assembling crosslinked actin network, fascin (CS-FSC01; Cytoskeleton Inc.), Fimbrin (Supplementary Note 3), and α-actinin (AT01; Cytoskeleton Inc.) were used at 1 μM, 1 μM, and 0.7 μM, respectively. In the experiments assembling branched and linear actin network, Arp2/3 (RP01P-B; Cytoskeleton Inc.), mDia1 (Supplementary Note 3), VCA (Supplementary Note 3), and profilin (PRO2; Cytoskeleton Inc.) were used at 0.3 μM, 0.3 μM, 1.5 μM, and 3 μM, respectively. In the experiments severing actin network, gelsolin (HPG6-A; Cytoskeleton Inc.) and cofilin (CF01-C; Cytoskeleton Inc.) were used at 56 nM and 6 μM, respectively. Details of protein purification and labeling procedures are available in the Supplementary Note 3.

### Preparation of water-in-oil droplets
Water-in-oil droplets were prepared by following the previously described method[98,99]. First, the mixture of the protein-NADH-coupled assay were prepared in a 0.65 mL tube. Separately, 10 μL of lipid-oil mixture was prepared in another 0.65 mL tube. Next, 0.5 μL of the protein-NADH-coupled assay mixture was added to the 10 μL of lipid-oil mixture in the same tube. The tube was gently tapped 2–3 times with a finger to generate water-in-oil emulsion droplets. Immediately after emulsification, the water-in-oil droplets were carefully transferred onto a coverslip mounted on the confocal microscope. All actin-associated proteins and enzymatic systems of NADH-coupled assay were kept on ice until use. It should be noted that some filaments can be nonspecifically absorbed to the water/oil interface due to the amphiphilic nature of actin. However, given the significantly small surface-to-volume ratio of the droplets, the influence of such nonspecifically adsorbed filaments on bulk actomyosin contraction and ATP consumption is negligible. It is important to note that we utilize PEG2000PE lipid to minimize the nonspecific adhesion of filaments to the water/oil interface. In some movies, as we capture the mid-plane of the spherical droplets, actomyosin network contraction also occurs in the z-direction, leading to a sudden appearance in some regions. The small round structures observed are partially coalesced emulsion droplets near the droplet interface, which cannot be entirely excluded in the current experimental system of water-in-oil emulsion.

### Microscopy
Images were acquired by Leica DMi8 inverted microscope equipped with a 63× 1.4-NA oil immersion lens (Leica Microsystems), a spinning-disk confocal (CSU22; Yokagawa), and sCMOS camera (Zyla; Andor Technology) controlled by Andor iQ3 (Andor Technology). The NADH fluorescence was imaged using a filter set consisting of 350 ± 50 nm excitation filter, 415 nm long-pass dichroic, and 460 ± 50 nm emission filter (Leica). Illumination from a mercury halide light source (11504120; Leica EL6000 Metal Halide) was used to excite the NADH in droplets. The excitation source was shuttered after every acquisition to minimize the photobleaching. The number of droplets recorded in each experiment is constrained by several factors. Firstly, to calculate the ATP consumption rate, we need to capture the time course of NADH fluorescence decay. Consequently, within a single experiment involving the preparation of one tube of water-in-oil emulsion, the focus is limited to the region of interest (ROI), despite having several hundred droplets outside the ROI. Secondly, we employ a relatively low ratio of the reaction solution to oil (1:20, solution:oil), ensuring

that droplets are adequately separated to prevent coalescence during measurements. Thirdly, we use high magnification objectives (63×), further constraining the size of the ROI in our confocal microscope. Due to these constraints, the measurement of the NADH fluorescence decay time course is limited to 1–3 droplets in a single experiment (Supplementary Fig. 1).

## Image analysis

The fluorescence intensity of NADH within a droplet was analyzed using Fiji/ImageJ (NIH), Microsoft Excel, and custom code written in MATLAB. The mean intensity within a circle with a half-radius of a droplet was determined and subtracted by the mean intensity of the background region. The photobleaching was corrected using exponential fitting method[100] for the droplets encapsulating NADH without any proteins. To relate the fluorescence intensity to the NADH concentration, calibration measurements were performed by adding known amounts of NADH (Supplementary Fig. 1). Because of the 1:1 stoichiometric ratio between ADP and NADH (Fig. 1c), the slope of the NADH fluorescence decay gives ATP consumption rate.

Quantitative image analysis on contracting actomyosin network was performed by using a custom code written in MATLAB. We measure the displacement vectors between individual frames at every spatial location $\mathbf{u}(\mathbf{r},t)$ to using a public domain Particle Image Velocimetry (PIV) program implemented as a Fiji/ImageJ plugin. Mean compressive strain $\varepsilon$ of the actin network is calculated from the divergence of the cumulative displacement field, $\varepsilon(t) = -\int_0^t dt' \int d\mathbf{r}[\nabla \cdot \mathbf{u}(\mathbf{r},t')]/\int d\mathbf{r}$. The spatial integral is taken over the negative divergence within a droplet to calculate the compressive strain. The subsequent change in cross-sectional area is calculated as $\Delta A(t) = A_0 + \int_0^t dt' \int d\mathbf{r}[\nabla \cdot \mathbf{u}(\mathbf{r},t')]$, where $A_0$ is the initial cross-sectional area of the network. The network radius $R$ was estimated by approximating the spherical symmetry of the contracting network and the cross-sectional area to be $\pi R^2$. Alternatively, we also compute cross-sectional area of the actin network using actin fluorescence image and confirmed the consistency with PIV results (Supplementary Fig. 6). The fluorescence intensity of actin was detected through the binarization of fluorescence images after applying median filter to remove the shot noise in the images taken by the confocal microscope.

## Statistical analysis

Statistical tests comparing distributions are done with the Wilcoxon rank sum test. All data displayed as a single value with an error bar is quoting the mean ± standard deviation. Fitted lines are shown to reject the null hypothesis to an extent that depends on the quoted $p$ value. The symbols *, **, and *** represent $p < 0.05$, 0.01, and 0.001, respectively.

## Reporting summary

Further information on research design is available in the Nature Portfolio Reporting Summary linked to this article.

## Data availability

Raw data supporting the finding of this manuscript are available from the corresponding author upon request because of the large size of the time lapse image data. The data generated in this study are provided in the Supplementary Information and Source Data file. Source data are provided with this paper.

## Code availability

Code supporting the findings of this manuscript are available from the corresponding authors upon request.

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

## Acknowledgements

The authors acknowledge funding ARO MURI W911NF-14-1-0403 and M.P.M, NIH RO1 GM126256 to M.P.M, NIH U54 CA209992 to M.P.M., and Human Frontier Science Program (HFSP) grant # RGY0073/2018 to M.P.M. M.P.M. and R.S. acknowledge support from Yale start-up funds. R.S. acknowledges support from the Overseas Postdoctoral Fellowships of the Uehara Memorial Foundation and the Overseas Research Fellowships of the Japan Society for the Promotion of Science (JSPS). Any opinion, findings, and conclusions or recommendations expressed in this material are those of the authors and do not necessarily reflect the views of the NSF, NIH, ARO, Royal Society, HFSP, Uehara Memorial Foundation, or JSPS.

## Author contributions

R.S. and M.P.M. designed research. R.S. contributed new regents/analytic tools. R.S. acquired the experimental data. R.S. analyzed the experimental data. R.S. drafted the paper. R.S. and M.P.M. edited the paper.

## Competing interests

The authors declare no competing interests.
