## [Peer Review File · Nature Communications]

Reviewers' Comments:

Reviewer #1:

Remarks to the Author:

The manuscript entitled "F-actin architecture determines conversion of chemical energy into mechanical work" by Ryota Sakamoto and Michael P. Murrell describes experiments with actomyosin networks reconstituted in oil droplets. For different network architectures, they estimated the ATP consumption and correlated it with the mechanical work produced by the network. Understanding the energy consumption of different actin architectures is a very important issue, although underestimated in the literature. In addition, how distinct F-actin architectures affect ATP consumption and energy consumption is still unclear. In this study, the authors established a quantitative link between actin network architecture and energy consumption. They demonstrate the influence of F-actin architecture on force generation and energy consumption.

Overall, the manuscript is well written and the experiments are well described. The test for measuring ATP consumption in real time in oil droplets is very interesting and could be widely used in the future in the field of reconstituted systems. However, I have a few major concerns that I would like the authors to address before I can recommend publication of the article in *Nature communications*. My concerns are about the definition and demonstration of dynamic steady state in Figure 4, the number of droplets analyzed in each condition and the various concentrations used in the different experiments that might change the conclusions of the paper.

Main concerns:

1. The assertions in Figure 4 are not justified and, above all, the notion of a dynamic steady state is not demonstrated. Do the authors mean to describe a contractile dynamic steady state as studied by Sonal et al. (*JCS* 2019) or a dynamic steady state based on polymerization/depolymerization/recycling as studied by Colin et al. (*EMBO Journal* 2023)? In the experiments shown by the authors in Figure 4 and Supplementary Figure 6, there is no demonstration that there is new actin polymerization after disassembly. To be able to claim that they have reconstituted a dynamic steady state, the authors would have to characterize it and, more specifically, show that there is actin turnover (i.e. several polymerization cycles). Otherwise, the conclusions drawn from the experiments in Figure 4 and Supplementary Figure 6 are not based on solid arguments.

In addition, in the discussion, the authors write: "the dynamic steady state without maintaining any structures consumed a larger amount of ATP". The authors need to clarify this unclear statement.

2. Statistics and number of droplets analyzed in each experiment. I am a bit surprised by the statistics: for 3 experiments only 10 droplets could be found? Usually, it is possible to generate hundreds of droplets in an experiment. So I wonder why the statistics only relate to 10 droplets? In addition, it is always the same example of droplet that is shown for "regular actin". It would be interesting to see a few different examples (images and movies) for each condition studied in the manuscript.

3. Concentrations and experimental conditions.

- ATP and NADH concentrations. The authors have used an initial ATP concentration of 5 mM, while the initial NADH concentration is 460 μ M. What is the reasoning behind these concentrations? Why the authors use such a high initial ATP concentration? Since the NADH concentration is only 460 μ M, the authors must have stopped their measurements when the NADH is fully consumed (which is not the time when all the ATP is consumed). It would also have been instructive to lower the ATP concentration, to see the effect on its rate of consumption and on the power generated by the network.

- VCA and Arp2/3 complex concentrations. The works of Bendix et al. (*Biophys. Journal* 2008) and Ennomani et al. (*Current Biology*, 2016) have shown that contractility depends on actin network connectivity. In Figure 3, the concentrations of VCA and Arp2/3 complex are very high, resulting in a very dense branched network (i.e with high connectivity) that could hinder myosin II contraction. It would be interesting for the authors to decrease the concentrations of VCA and Arp2/3 in order to reduce the density of the network and see the effect on ATP consumption and mechanical power.

Minor points:

1. How the oil droplets are generated? What is their average size? Does the size of the droplets

- influence the energy consumption? This should be mentioned somewhere in the manuscript.
2. The authors used a polymerization buffer with dextran and sucrose instead of the classic methylcellulose used in reconstituted systems. Could they explain why?
 3. Figure 1e/1f: there appears to be a non-specific interaction of actin filaments with the droplet surface. Could the authors explain this? On the Movie S1, in the examples for 12.5 nM and 25 nM myosin, it seems that these filaments emerge from the droplet surface (forming round structures?), could the authors comment on this point?
 4. There is a typo in the title of Figure 3 "Blanched" instead of "Branched"

Reviewer #2:

Remarks to the Author:

Overview: This manuscript investigated the influence of different actin network architectures by actin binding proteins on their contractility and ATP consumption. Myosin motors work in tandem with the actin cytoskeleton to create contractile forces within the cell powered by ATP. The actin cytoskeleton arranges itself in a variety of network architectures that can be utilized for different functions, depending on the actin binding protein used. The authors aimed to investigate how the different actin architectures influences myosin energy consumption and conversion efficiency. They used an NADP-coupled assay to investigate the ATP consumption rates of different reconstituted actin systems. Then they employed imaging to investigate the mechanical work each system was able to produce.

Overall, this work is very well thought out and the experimental designs are thorough to determine the ATP consumption and power produced by each of their conditions. The results of this work may be of significance to provide valuable insights into the mechanism of actomyosin contractility, and the effects of various actin binding proteins on those systems. Below are some points that may help improve the manuscript.

Major Comments:

1. Estimations of the mechanical work performed by myosin were calculated using the change in volume of the water-in-oil drops. Did addition of any of the actin binding proteins and their resulting architectures alter the average shape of the droplets, or cause deformations as they might in the cell? Also, can you comment on how the results obtained in your 2D droplet may differ if they were measured inside a 3D cell environment?
2. The authors explored the use of actin bundling proteins and actin nucleators based architectures on myosin contractility. The cell uses different actin binding proteins for different locations within the cell, such as the filopodia protruding from the lamellipodia. Can the authors comment on how a combination of actin branching and bundling may influence myosin contraction, such as in the case of filopodia production.
3. It would be interesting to know if measurements of the rates of ATP consumption of branched, bundled, and crosslinked actin without myosin present, were performed to see if their consumption rates were comparable? Differences in the ATP consumption rate of varying F-actin network architecture may also contribute to the differences observed between them.
4. Can you comment on the rigidity of each of the actin architecture. Does the binding of the varying actin binding proteins used in this study change the actin conformation enough to alter filament rigidity and resulting myosin contractility.
5. It would be useful to explain the choice of specific crosslinker concentrations. For example, why was the α -actinin concentration (0.7 μ M) chosen to be lower than the concentrations used for fimbrin and fascin (1 μ M)? Related to that, why were such different concentrations used for the actin severing proteins, cofilin (6 μ M) and gelsolin (56 nM)?

Minor Comments:

1. The discussion is very thorough but could use some subheadings to differentiate the different topics explored within that section and better direct the reader on what each section is explaining.
2. Can you address the changes which may occur for myosin-induced mechanical work, strain, and power when physiological conditions, such as macromolecular crowding and protein-protein interactions, are introduced?
3. Figure 3, should be "Branched" instead of "Blanched"

4. In line 228, 6 μM cofilin;[Actin]:[Gelsolin] should be corrected to [Actin]:[Cofilin].

Reviewer #3:

Remarks to the Author:

The manuscript by Sakamoto and Murrell describes various experiments to probe energy conversion in actin-myosin networks. Specifically, they reconstitute these gels inside oil-in-water droplet using various molecular controls (cross-linkers, gelsolin, cofilin, etc.) to tune architectural properties of the network. By using an NADH-based assay they determine ATP hydrolysis, and the mechanical work is estimated based on imaging data and assumptions.

Understanding how chemical energy that is consumed at the level of individual molecular motors with each hydrolysis-cycle is converted into mechanical work that leads to networks deformations is timely, interesting, and especially determining the role of network architecture in this makes this work relevant to several mechanical processes in cells. However, in its current form my support for publishing this manuscript is limited by my concerns about the methods the authors use to determine the work and power generated in these networks, on which the central claims of this paper rest. I will outline this point, together with several other concerns and questions below.

1. The authors estimate the work performed by myosin (and the power), by determining changes in the volume of the network as it contracts which is multiplied by an estimate of the stress. I am concerned that this is a very crude estimate of the work, and little or no evidence is provided as to how accurate this estimate could be. A first aspect that is missing is the fraction energy that is stored in volume conserving modes of deformations of the network, which is not discussed in the text (as an extreme case, imagine anchoring the network to a rigid boundary preventing contraction, implying that all energy goes into volume-conserving modes of deformation). This fraction will be sensitive to the nonlinear elasticity of the network and will likely also dependent on network architecture, which means that even if the work performance would not change with architecture the estimate used by the authors could change. A second aspect that is concerning is the simple argument that is used to estimate the stress production by the myosin motors. Here to, I would expect (possibly large) corrections that are sensitive to network architecture. Finally, a third aspect is that not all work is stored is likely stored in elastic energy. There are dissipative effects such as plastic reorganization and fracture (Wollrab et al. Journal of cell science 132, no. 4 (2019): jcs219717.), which could be significant.

The authors do mention that their estimate is a bound, but it is unclear how tight the bound is and is unclear how the tightness of the bound varies with conditions, which makes it difficult to use it to compare systems under different conditions. Unfortunately, no direct experimental evidence is given to show how accurate the work estimate is.

Given these issues, I am concerned about presenting these crude estimates in main figures of the paper, and I am concerned that the work and power estimates are used as central observables and arguments that lead to the main conclusions of the paper.

2. An NADH assay is used to estimate energy consumption. One concern I have with this approach is that not all ATP hydrolysis by myosin necessarily results in force-generation in the network, and this might also depend on network architecture. How the authors performed controls to investigate how much of the ATP consumption results can be effectively transferred to the actin network?

3. Fig 1n shows energy input versus mechanical output and the authors argue that this relation is linear. Given the limited number of data points (only 3) and large error bar on the 50 nM case, I think the evidence is thin. More (precise) data would be needed to draw this conclusion.

4. A separate question tangentially related to this figure: What percentage of the energy input is converted into work, and how does this efficiency depend on network architecture? The line about "how distinct F-actin architectures impact ATP consumption and energy conversion efficiency" made me expect results on this topic.

5. The authors present strain fields in various maintext figures, but they are not discussed in detail. Can they expand on the presence of both large compressive and extensile strains and the homogeneous structure of these strain fields?

6. Some relevant context in the discussion is missing. A relevant pioneering paper is not mentioned Bendix et al. Biophysical journal 94, no. 8 (2008): 3126-3136. Also, I want to bring to the attention: Nitta et al. Nature materials 20, no. 8 (2021): 1149-1155 and Jia et al. Nature Materials 21, no. 6 (2022): 703-709. The latter specifically discussed work production in actin/myosin networks.

Response Letter
F-actin architecture determines the conversion of chemical energy into mechanical work
R. Sakamoto & M.P. Murrell

We greatly appreciate all the reviewers for carefully reading our manuscript and for their insightful comments. In light of their suggestions, we have made changes to the manuscript and responded to each comment below.

Response to Reviewer #1

The manuscript entitled “F-actin architecture determines conversion of chemical energy into mechanical work” by Ryota Sakamoto and Michael P. Murrell describes experiments with actomyosin networks reconstituted in oil droplets. For different network architectures, they estimated the ATP consumption and correlated it with the mechanical work produced by the network. Understanding the energy consumption of different actin architectures is a very important issue, although underestimated in the literature. In addition, how distinct F-actin architectures affect ATP consumption and energy consumption is still unclear. In this study, the authors established a quantitative link between actin network architecture and energy consumption. They demonstrate the influence of F-actin architecture on force generation and energy consumption.

Overall, the manuscript is well written and the experiments are well described. The test for measuring ATP consumption in real time in oil droplets is very interesting and could be widely used in the future in the field of reconstituted systems. However, I have a few major concerns that I would like the authors to address before I can recommend publication of the article in Nature communications. My concerns are about the definition and demonstration of dynamic steady state in Figure 4, the number of droplets analyzed in each condition and the various concentrations used in the different experiments that might change the conclusions of the paper.

We are grateful to the reviewer for identifying our work is very interesting and could be widely used in the future in the field of reconstituted systems. We agree that some experimental conditions were not clarified enough in the original manuscript. We thus made the following revisions:

Main concerns:

1. The assertions in Figure 4 are not justified and, above all, the notion of a dynamic steady state is not demonstrated. Do the authors mean to describe a contractile dynamic steady state as studied by Sonal et al. (JCS 2019) or a dynamic steady state based on polymerization/depolymerization/recycling as studied by Colin et al. (EMBO Journal 2023)? In the experiments shown by the authors in Figure 4 and Supplementary Figure 6, there is no demonstration that there is new actin polymerization after disassembly. To be able to claim that they have reconstituted a dynamic steady state, the authors would have to characterize it and, more specifically, show that there is actin turnover (i.e. several polymerization cycles).

Otherwise, the conclusions drawn from the experiments in Figure 4 and Supplementary Figure 6 are not based on solid arguments.

In addition, in the discussion, the authors write: “the dynamic steady state without maintaining any structures consumed a larger amount of ATP”. The authors need to clarify this unclear statement.

We appreciate this insightful comment. We acknowledge that the definition of the dynamic steady state was not sufficiently clear in the initial manuscript, which may have caused confusion. It is important to clarify that when we refer to the 'dynamic steady-state' in our study, we specifically focus on the propensity to retain a constant F-actin density and not condense (contract). In our context, the term 'dynamic steady-state' describes a state in which there are no contracted actomyosin aggregates present. This is crucial because the presence of contracted actomyosin increases local actin density, motion of myosin is confined, potentially influencing the load-dependent ATP hydrolysis rate of myosin.

To clarify this point, we define the 'dynamic steady-state' as a state that satisfies the following conditions: (i) the macroscopic strain rate of the actin network is zero ($\dot{\epsilon} \equiv -\langle \nabla \cdot \mathbf{v} \rangle = 0$) and (ii) the spatially averaged magnitude of velocity takes a finite value ($\langle |\mathbf{v}| \rangle > 0$), as outlined in our previous study (Seara et al., *Nat. Commun.* **9**, 4948 (2018)). We demonstrated that the gelsolin-severed system exhibits a strain rate fluctuating around zero with a finite magnitude of velocity (**Additional Supplementary Fig. 11**). Therefore, the gelsolin-severed actomyosin system can be characterized as being in a dynamic-steady state. By contrast, when the network is stabilized by phalloidin, the strain rate takes a large value due to the presence of macroscopic contractile flow. Consequently, the phalloidin-stabilized actomyosin system does not satisfy the dynamic steady-state condition. To address the clarification of the dynamic steady-state definition in our study, we have included this point in the Results section (**pp. 9, lines 267-280**), modified the dynamic steady-state condition in **Fig. 4C**, and added a **Supplementary Fig. 11**.

Notably, our results in Supplementary Fig. 12 demonstrate that in the absence of the entangled F-actin network, myosin motors bound to F-actin consume ATP, exhibiting dynamic steady-state of myosin motion without forming actin aggregates. The observed higher ATP consumption rate in the presence of gelsolin compared to the control (entangled network) support the relevance of our conclusions in Fig. 4. We have revised the unclear statement “*Furthermore, the dynamic steady state without maintaining any structures consumed a larger amount of ATP.*” in the manuscript as “*Furthermore, maintaining the dynamic steady-state of myosin motion without confining myosin within actin aggregates consumed a larger amount of ATP.*” (**pp. 11, lines 334-335**).

Regarding the reviewer's comments on actin turnover, we would like to emphasize that, for our experimental focus on the dynamic steady state of myosin motion, this condition is not crucial. Our primary interest lies in understanding the ATP consumption when myosin motion

is not constrained by contracted actin aggregates. Nevertheless, we confirmed that the stabilization of F-actin by phalloidin slows the ATP consumption rate of myosin (Supplementary Fig. 14). This suggests that preventing actin turnover may accumulate more load on myosin, thereby slowing down its ATP hydrolysis rate. It would be interesting for future studies to systematically examine the influence of actin turnover on energy consumption.

Additional Supplementary Figure 11 : The actomyosin system severed by gelsolin is in a dynamic steady-state. (A and B) PIV analysis on actomyosin network contraction in the presence of 56 nM Gelsolin (A) or 17 μM Phalloidin (B). Yellow vectors are the total displacement \mathbf{u} over 20 min and vector magnitudes are normalized by the maximum displacement (left). The colormap represents local strain fields (right). (C) Macroscopic strain rate ($\dot{\epsilon} \equiv -\langle \nabla \cdot \mathbf{v} \rangle$) calculated from the instantaneous velocity field \mathbf{v} between each frame (n=9 droplets and N=8 independent experiments in Gelsolin; n=8 and N=5 in Phalloidin). (D) Spatially averaged magnitude of velocity ($\langle |\mathbf{v}| \rangle$) (n=9 droplets and N=8 independent experiments in Gelsolin; n=8 and N=5 in Phalloidin). Scale bars, 10 μm .

2. *Statistics and number of droplets analyzed in each experiment. I am a bit surprised by the statistics: for 3 experiments only 10 droplets could be found? Usually, it is possible to generate hundreds of droplets in an experiment. So I wonder why the statistics only relate to 10 droplets? In addition, it is always the same example of droplet that is shown for “regular actin”. It would be interesting to see a few different examples (images and movies) for each condition studied in the manuscript.*

We appreciate this comment. The number of droplets recorded in each experiment is constrained by several factors. First, to calculate the ATP consumption rate, we need to

capture the time course of NADH fluorescence decay. Consequently, within a single experiment involving the preparation of one tube of water-in-oil emulsion, the focus is limited to the region of interest (ROI), despite having several hundred droplets outside the ROI (e.g., n=558 droplets were analyzed for the calculation of calibration curve in Supplementary Fig. 1). Second, we employ a relatively low ratio of the reaction solution to oil (1:20, solution:oil), ensuring that droplets are adequately separated to prevent coalescence during measurements. Third, we use high magnification objectives (63x), further constraining the size of the ROI in our confocal microscope. Due to these constraints, the measurement of the NADH fluorescence decay time course is limited to 1-3 droplets in a single experiment (**Revised Supplementary Fig. 1A**). We have incorporated an explanation of these points in the Method section (pp. 17, lines 542-550) and included a representative entire field of view image in **Revised Supplementary Fig. 1A**. Additionally, different examples of droplets for each condition (varied myosin concentration, different crosslinkers, different nucleators, in which three different examples of controls and one different examples for each condition) are included in the Supplementary Movies (**Movies S2, S4, and S6**).

Revised Supplementary Figure 1 : (A) The entire non-cropped field of view in confocal microscopy. The single region of interest typically contains 1-3 droplets, which ensures that droplets are adequately separated to prevent coalescence during measurements. Scale bar, 20 μm. (B) Droplets containing known concentrations of NADH were analyzed (n=558 droplets and N=6 independent experiments). Data points are mean+/-SD. The dashed line is the linear fitting, $I_{NADH} = 0.1173 \times [NADH]$.

3. Concentrations and experimental conditions. – ATP and NADH concentrations. The authors have used an initial ATP concentration of 5 mM, while the initial NADH concentration is 460 μM. What is the reasoning behind these concentrations? Why the authors use such a high initial ATP concentration? Since the NADH concentration is only 460 μM, the authors must have stopped their measurements when the NADH is fully consumed (which is not the time when all the ATP is consumed). It would also have been instructive to lower the ATP

concentration, to see the effect on its rate of consumption and on the power generated by the network.

Thank you for this comment. The ATP concentration has been optimized to achieve highly active actin polymerization and myosin contraction in liposome systems, as demonstrated in our group (Murrell et al., *Biophys. J.* **100**, 1400 (2011); Sakamoto et al., *Commun. Biol.* **6**, 325 (2023)). It is important to note that the ATP concentration is not a critical factor for the current NADH assay. Since the consumed ADP is continually regenerated through the PK/PEP system, the ATP concentration remains constant until when all NADH is consumed. Following the reviewer's suggestion, we performed an additional experiment with a lower ATP concentration of 1 mM (**Additional Supplementary Fig. 7 and Movie S4**). The reduced ATP concentration resulted in a slightly decelerated ATP consumption rate and mechanical power. This result aligns with reports indicating that the myosin II sliding velocity in muscle fibers and in *in vitro* motility assays increases proportionally to ATP concentration (Cooke & Bialek, *Biophys. J.* **28**, 241 (1979); Kron & Spudich, *PNAS* **83**, 6272 (1986)). At low ATP concentration, the network may exhibit increased transient crosslinking due to the higher affinity of ADP-myosin for F-actin (Köhler et al., *PLoS ONE* **6**, e23798 (2011)), which could also contribute to load-dependent ATP hydrolysis. We mentioned this additional experiment in the Results section (**pp. 6, lines 159-164**).

In contrast, the choice of NADH concentration is based on technical considerations. First, we selected a sufficiently low NADH concentration to guarantee the complete decay of NADH fluorescence in ~20 minutes at [Myosin] = 50 nM. This enhances the visibility and clarity in identifying ATP consumption. More importantly, as the conversion of NADH requires a comparable amount of PEP and the associated concentrations of PK and LDH, using a larger NADH concentration would significantly increase the overall concentration of the additional enzymatic system. To minimize the impact of NADH assay-related enzymatic systems on the conventional actin polymerization buffer, we selected a low NADH concentration for our measurements. An additional note has been added to the Method section to clarify the reasoning behind the choice of NADH concentration (**pp. 16, lines 494-500**).

Additional Supplementary Figure 7 : ATP concentration dependence of the ATP consumption and mechanical power. (A and B) Time-lapse images showing the contraction of the actin network with ATP concentration 5 mM (A) and 1 mM (B) (actin in red, myosin in green), and the NADH fluorescence (in white). (C) Instantaneous power over time ($n=29$ droplets and $N=20$ independent experiments in $[ATP] = 5$ mM; $n=20$ and $N=13$ in $[ATP] = 1$ mM). Inset shows mean compressive strain over time. Curves are mean \pm std. (D) Boxplot showing the maximum instantaneous power performed by myosin extracted from (C). (E) NADH fluorescence over time normalized by the initial time point ($n=23$ and $N=11$ in $[ATP] = 5$ mM; $n=19$ and $N=10$ in $[ATP] = 1$ mM). (F) Boxplot showing the ATP consumption rate in (E). * represents $p < 0.05$. Scale bars, 10 μ m.

4. Concentrations and experimental conditions. - VCA and Arp2/3 complex concentrations. The works of Bendix et al. (Biophys. Journal 2008) and Ennomani et al. (Current Biology, 2016) have shown that contractility depends on actin network connectivity. In Figure 3, the concentrations of VCA and Arp2/3 complex are very high, resulting in a very dense branched network (i.e with high connectivity) that could hinder myosin II contraction. It would be interesting for the authors to decrease the concentrations of VCA and Arp2/3 in order to reduce the density of the network and see the effect on ATP consumption and mechanical power.

We appreciate this comment. Our primary focus is on exploring the influence of highly branched F-actin architecture, and the concentrations of Arp2/3 and VCA have been meticulously optimized in prior studies to attain the highly branched architecture in our group (Murrell et al., *Biophys. J.* **100**, 1400 (2011); Muresan et al., *Nat. Commun.* **13**, 7008 (2022); Sakamoto et al., *Commun. Biol.* **6**, 325 (2023)). It is worth noting that the cellular Arp2/3

concentration is on the order of μM ; thus, using the high Arp2/3 concentration is also relevant in the biological context (Cao et al., *Nat. Cell Biol.* **22**, 803 (2020)).

Following the reviewer's suggestion, we performed additional experiments with lower Arp2/3 concentrations (**Additional Supplementary Fig. 10 and Movie S10**). In the main text, we specified the Arp2/3 concentration as 300 nM ($[\text{Arp2/3}]/[\text{Actin}]=0.05$). In the supplementary experiment, we reduced the Arp2/3 concentration to 60 nM ($[\text{Arp2/3}]/[\text{Actin}]=0.01$) and 6 nM ($[\text{Arp2/3}]/[\text{Actin}]=0.001$). Notably, even at 60 nM Arp2/3, network contraction was effectively prevented while maintaining an ATP consumption rate as fast as that without Arp2/3. By contrast, we observed network contraction at 6 nM Arp2/3, where mechanical power was lower than that without Arp2/3, along with a slowed ATP consumption rate compared to 60 nM Arp2/3 and the condition without Arp2/3. This result indicates that at a lower concentration of Arp2/3, Arp2/3 branching partially functions as a crosslinker, inducing load-dependent ATP hydrolysis of myosin within contracted networks. Together, these results suggest that the degree of branching via Arp2/3 is crucial in regulating contractile behavior and myosin ATP consumption. We have incorporated this additional experiment in the Results section (**pp. 8, lines 232-239**).

Additional Supplementary Figure 10 : Arp2/3 concentration dependence of the ATP consumption and mechanical power. (A) Snapshots showing the contraction of the actin network without Arp2/3, with 60 nM Arp2/3, and with 6 nM Arp2/3 (left, actin in red, myosin in green), and the NADH fluorescence (right, in white). (B) Instantaneous power over time ($n=12$ droplets and $N=11$ independent experiments in without Arp2/3; $n=9$ and $N=6$ in $[\text{Arp2/3}] = 60 \text{ nM}$; $n=12$ and $N=7$ in $[\text{Arp2/3}] = 6 \text{ nM}$). Inset shows mean compressive strain over time. Curves are mean \pm std. (C) Boxplot showing the maximum

instantaneous power performed by myosin extracted from (B). (D) NADH fluorescence over time normalized by the initial time point (n=8 droplets and N=6 independent experiments in without Arp2/3; n=7 and N=5 in [Arp2/3] = 60 nM; n=8 and N=5 in [Arp2/3] = 6 nM). (E) Boxplot showing the ATP consumption rate in (D). *, **, and *** represent $p < 0.05$, $p < 0.01$, and $p < 0.001$, respectively. n.s., not significant. Scale bars, 10 μm .

Minor points:

1. How are the oil droplets generated? What is their average size? Does the size of the droplets influence the energy consumption? This should be mentioned somewhere in the manuscript.

We appreciate this suggestion. The droplets were generated by following the previously developed procedure (Sakamoto et al., *Nat. Commun.* **11**, 3063 (2020)). First, the mixture of the protein-NADH-coupled assay were prepared in a 0.65 mL tube. Separately, 10 μL of lipid-oil mixture was prepared in another 0.65 mL tube. Next, 0.5 μL of the protein-NADH-assay mixture was added to the 10 μL of lipid-oil mixture in the same tube. The tube was gently tapped 2-3 times with a finger to generate water-in-oil emulsion droplets. Immediately after emulsification, the water-in-oil droplets were carefully transferred onto a coverslip mounted on the confocal microscope. All actin-associated proteins and enzymatic systems of NADH-coupled assay were kept on ice until use. We have included the preparation procedure of droplets in the Method section in the main text (pp. 16, lines 519-534).

The measurements of ATP consumption were performed for the droplets within the comparable range of droplet size for each condition to reduce the variability. The additional data of the boxplot shows that the range of the droplet size was comparable in different conditions, and the size-dependence of the ATP consumption rates was not significant (Additional Supplementary Figure 2). These additional data have been included in Supplementary Fig. 2.

Additional Supplementary Figure 2 : The analyzed droplet size was comparable in different conditions. (A) Boxplot showing the droplet size analyzed for ATP consumption rates at different myosin concentration (n=12 droplets and N=6 independent experiments for 12.5 nM; n=9 and N=5 for 25 nM; n=15 and N=8 for 50 nM). (B) Boxplot showing the droplet size analyzed for ATP consumption rates in various crosslinkers (n=9 and N=4 in without crosslinkers; n=5 and N=3 in fascin; n=6 and N=3 in fimbrin; n=8 and N=4 in α-actinin). (C) Boxplot showing the droplet size analyzed for ATP consumption rates in various nucleator compositions (n=9 and N=5 in without nucleators; n=8 and N=5 for Arp2/3 only; n=8 and N=4 in mDia1 only; n=10 and N=5 in [Arp2/3]:[mDia1] = 1:1; n=8 and N=4 in [Arp2/3]:[mDia1] = 1:0.1). (D) Scatter plot showing the droplet size vs ATP consumption rates at different myosin concentrations. (E) Scatter plot showing the droplet size vs ATP consumption rates in various crosslinkers. (F) Scatter plot showing the droplet size vs ATP consumption rates in various nucleator compositions. *: $p < 0.05$. n.s.: not significant.

2. The authors used a polymerization buffer with dextran and sucrose instead of the classic methylcellulose used in reconstituted systems. Could they explain why?

The inclusion of dextran and sucrose is a result of previously optimized conditions to achieve highly active actin polymerization and myosin contraction in liposome systems, as established in our group (Murrell et al., *Biophys. J.* **100**, 1400 (2011); Sakamoto et al., *Commun. Biol.* **6**, 325 (2023)). It is important to note that unlike methylcellulose, dextran does not cause bundling, and thus, its impact on F-actin architecture is negligible. We have chosen the buffer suitable for liposome systems because we plan to extend the current system to liposomes, allowing for the simultaneous measurement of significant membrane deformation and ATP

consumption. We referred this point in the Method section (pp. 15, lines 480, 482-483).

3. *Figure 1e/1f: there appears to be a non-specific interaction of actin filaments with the droplet surface. Could the authors explain this? On the Movie S1, in the examples for 12.5 nM and 25 nM myosin, it seems that these filaments emerge from the droplet surface (forming round structures?), could the authors comment on this point?*

Due to the amphiphilic nature of actin, some filaments can be nonspecifically absorbed to the water/oil interface. However, given the significantly small surface-to-volume ratio of the droplets, the influence of such nonspecifically adsorbed filaments on bulk actomyosin contraction and ATP consumption is negligible. It is important to note that we utilize PEG2000PE lipid to minimize the nonspecific adhesion of filaments to the water/oil interface. In some movies, as we capture the mid-plane of the spherical droplets, actomyosin network contraction also occurs in the z-direction, leading to a sudden appearance in some regions. The small round structures observed are partially coalesced emulsion droplets near the droplet interface, which cannot be entirely excluded in the current experimental system of water-in-oil emulsion. We replaced the example of 25 nM myosin without a round structure present in **Movie S1**, and added different examples of movies with varied myosin concentrations (**Movie S2**). We clarified these points by adding an explanation in the Method section (pp. 17, lines 525-534).

4. *There is a typo in the title of Figure 3 “Blanched” instead of “Branched”.*

We have corrected this typo.

Response to Reviewer #2

Overview: This manuscript investigated the influence of different actin network architectures by actin binding proteins on their contractility and ATP consumption. Myosin motors work in tandem with the actin cytoskeleton to create contractile forces within the cell powered by ATP. The actin cytoskeleton arranges itself in a variety of network architectures that can be utilized for different functions, depending on the actin binding protein used. The authors aimed to investigate how the different actin architectures influences myosin energy consumption and conversion efficiency. They used an NADH-coupled assay to investigate the ATP consumption rates of different reconstituted actin systems. Then they employed imaging to investigate the mechanical work each system was able to produce.

Overall, this work is very well thought out and the experimental designs are thorough to determine the ATP consumption and power produced by each of their conditions. The results of this work may be of significance to provide valuable insights into the mechanism of

actomyosin contractility, and the effects of various actin binding proteins on those systems. Below are some points that may help improve the manuscript.

We are grateful to the reviewer for identifying our work is very thought out and of significance to provide valuable insights into the mechanism of actomyosin contractility, and the effects of various actin binding proteins on those systems.

Major comments:

1. Estimations of the mechanical work performed by myosin were calculated using the change in volume of the water-in-oil drops. Did addition of any of the actin binding proteins and their resulting architectures alter the average shape of the droplets, or cause deformations as they might in the cell? Also, can you comment on how the results obtained in your 2D droplet may differ if they were measured inside a 3D cell environment?

We appreciate this comment. Given that changes in droplet shape may introduce uncertainty in ATP consumption measurements, in this work, we focus the contraction of the volume network without inducing changes in droplet shape. To achieve this, we employed PEG2000PE lipid to prevent F-actin adhesion to the water/oil interface of the droplets, ensuring no observable shape changes due to actomyosin contraction. While the shape change is beyond the scope of this work, it would be a very interesting future direction to quantify membrane deformation and energy consumption simultaneously, mimicking the processes observed in cells.

It should be noted that our measurements were performed in 3D droplets, focusing on their mid-planes. Regarding the comment “*how the results obtained in your 2D droplet may differ if they were measured inside a 3D cell environment?*”, as the reviewer mentioned below, the macromolecular crowding and protein-protein interactions in physiological conditions could induce differences in results between droplets and cells. Given that macromolecular crowding may increase the viscosity of the solution, the contraction of the actomyosin network could be slowed by larger fluid drag. Additionally, protein-protein interactions introduce greater friction force counteracting the direction of actomyosin contraction. Consequently, we anticipate that the resulting myosin-induced mechanical power will be smaller than our measured values. We added these potential differences between droplet systems and cellular environment in the Discussion section (**pp. 14, lines 444-453**).

Additionally, one aspect that distinguishes the present system from cells is the deformation of the membrane, which underlies many biological processes from cell division to migration. In cells, the membrane-bound actomyosin networks perform mechanical work on the membrane to deform it. To recapitulate these more complex cellular behaviors, we plan to extend our ATP consumption measurements to liposomes in future studies, where significant membrane deformation can be induced by actomyosin contraction.

2. The authors explored the use of actin bundling proteins and actin nucleators based architectures on myosin contractility. The cell uses different actin binding proteins for different locations within the cell, such as the filopodia protruding from the lamellipodia. Can the authors comment on how a combination of actin branching and bundling may influence myosin contraction, such as in the case of filopodia production.

Thank you for this insightful comment. As a mechanism of filopodia production, the 'convergent elongation model' has been proposed (Svitkina et al., *J. Cell Biol.* **160**, 409 (2003)). In this model, the branched F-actin network serves as a scaffold for the protrusion of filopodia from lamellipodia. Filopodia are composed of parallel actin bundles crosslinked by fascin (Vignjevic et al., *J. Cell Biol.* **174**, 863 (2006)). Thus, myosin motors in lamellipodia could be exposed to a structural transition from a branching to a bundling architecture.

Our study demonstrated that the Arp2/3 network impedes myosin contraction, whereas fascin-crosslinked parallel bundles facilitate myosin contraction. Consequently, while myosin motors in lamellipodia contracts moderately, upon reaching filopodia, they may induce rapid contraction of filopodia. Thus, the structural shift from branched to bundled networks could be crucial in the sequential processes observed in adherent migratory cells: extending the leading edge, forming substrate adhesion, and rapidly contracting filopodia (Nemethova et al., *J. Cell Biol.* **180**, 1233 (2008)). We have incorporated this point into the Discussion section (**pp. 14, lines 413-423**), which have improved the biological implications of the manuscript.

3. It would be interesting to know if measurements of the rates of ATP consumption of branched, bundled, and crosslinked actin without myosin present, were performed to see if their consumption rates were comparable? Differences in the ATP consumption rate of varying F-actin network architecture may also contribute to the differences observed between them.

We are thankful to this comment. The original manuscript included the results of ATP consumption rates for branched and linear networks without myosin present, which were found to be comparable to the case of spontaneously polymerized actin (Supplementary Fig. 3). This indicates that the alteration of F-actin architecture does not significantly affect the ATP consumption of F-actin. Following the reviewer's suggestion, we additionally performed an experiment to measure the ATP consumption rates of crosslinked actin networks. The ATP consumption rates of crosslinked actin networks were comparable to that of actin networks without crosslinkers (**Additional Supplementary Fig. 9, Movie S7**). Therefore, the variation in ATP consumption rates with different F-actin architectures in the presence of myosin is predominantly attributable to myosin-based ATP consumption in the present experimental condition. We mentioned this point the Results section (**pp. 7, lines 204-213**) and included the additional data in **Supplementary Fig. 9, Movies S7 and S11**.

Additional Supplementary Figure 9 : ATP consumption rate of actin in the presence of actin crosslinkers. (A and B) Snapshots showing the actin fluorescence (A) and NADH fluorescence (B) within droplets containing actin and actin crosslinkers, fascin, fimbrin, or α -actinin without myosin present. (C) Normalized NADH fluorescence over time (n=11 droplets and N=5 independent experiments in control; n=8 and N=4 in w/o crosslinkers; n=6 and N=3 in fascin; n=7 and N=3 in fimbrin; n=9 and N=3 in α -actinin). Curves are mean \pm std. (D) Boxplot showing the ATP consumption rate in the presence of different actin crosslinkers (n=8 and N=4 in w/o crosslinkers; n=6 and N=3 in fascin; n=7 and N=3 in fimbrin; n=9 and N=3 in α -actinin). n.s., not significant. Scale bars are 10 μm .

4. Can you comment on the rigidity of each of the actin architecture. Does the binding of the varying actin binding proteins used in this study change the actin conformation enough to alter filament rigidity and resulting myosin contractility.

We acknowledge that rigidity of actin networks may change with architecture, as previously reported by rheological measurements (the elastic modulus of the Arp2/3-nucleated networks ~ 1 kPa (Marcy et al., *PNAS* **101**, 5992 (2004)), actin networks ~ 10 Pa; fascin-crosslinked networks ~ 50 Pa; α -actinin-crosslinked networks ~ 100 Pa (Tseng et al., *J. Biol. Chem.* **279**, 1819 (2004)); fimbrin-crosslinked network ~ 50 Pa (Klein et al., *Structure* **12**, 999 (2004)); note that elastic modulus of the crosslinked networks were measured at ~ 1 mol% crosslinkers to actin, so that the absolute elasticity can be much higher in the present system). The more rigid

the filaments are, the less bent they become, and mechanical work is more linearly applied during contraction, allowing to neglect nonlinear effects such as filament bending and fracture. We have incorporated these points in **Supplementary Note 2** to clarify the contribution of the rigidity of actin networks.

It is important to note that our focus is on how the F-actin architecture alters myosin contractility, rather than the rigidity of the filament itself. Specifically, we aim to compare branched and linear F-actin architecture nucleated by Arp2/3 and mDia1. For this purpose, the nucleator concentrations were chosen to be high enough to alter the contractile behavior of the actomyosin network as reported in the previous study (Muresan et al., *Nat. Commun.* **13**, 7008 (2022)). In the case of crosslinkers, architectural differences arise from the polarity of actin filaments in bundles (fascin is parallel; fimbrin is antiparallel) or filament spacing (fimbrin ~8 nm; α -actinin ~35 nm), not the difference in filament rigidity. For this purpose, crosslinker concentrations were chosen to be high enough to alter the motion of myosin thick filaments (Weirich et al., *Biophys. J.* **120**, 1957 (2021)).

Therefore, we have chosen the high enough concentrations of the crosslinkers and nucleators to impact myosin force generation via F-actin architecture. Investigating the effect of filament rigidity on myosin ATP consumption behavior would be an interesting future topic. We have clarified these points in the main text (**pp. 7, lines 186-187; pp. 12, lines 361-363**).

5. *It would be useful to explain the choice of specific crosslinker concentrations. For example, why was the α -actinin concentration (0.7 μ M) chosen to be lower than the concentrations used for fimbrin and fascin (1 μ M)? Related to that, why were such different concentrations used for the actin severing proteins, cofilin (6 μ M) and gelsolin (56 nM)?*

We chose the concentration of fimbrin and fascin to be high enough to observe the bundles. The slightly lower concentration of α -actinin compared to fascin and fimbrin is due to a technical reason that the original stock concentration of α -actinin was 10 μ M, while fascin and fimbrin were at 70 μ M. More importantly, it should be noted that the fascin, fimbrin, and α -actinin concentrations were high enough (5-10 mol% relative to actin) to impact the myosin force generation as demonstrated in previous studies (Weirich et al., *Biophys. J.* **120**, 1957 (2021)). Therefore, we believe that the present concentration choices are adequately high enough for investigating the impact of F-actin architecture on ATP consumption rates of myosin. We have incorporated this point into the main text to justify the choice of crosslinker concentrations (**pp. 7, lines 186-187; pp. 12, lines 361-363**).

Regarding severing proteins, our goal is to eliminate all actin aggregates that constrain myosin motion, and the concentration of severing proteins has been chosen to be high enough to eliminate actomyosin aggregates. To achieve this, we chose a cofilin concentration comparable to that of actin, as demonstrated by the previous study that the influence of cofilin on actin turnover saturates at [Actin]:[Cofilin]=1:1 (McCall et al., *PNAS* **116**, 12629 (2019)).

On the other hand, even at a low concentration of gelsolin (56 nM), it completely severs the filaments, allowing myosin to freely diffuse over space without its motion being constrained by actin aggregates (Supplementary Fig. 12). As we can prevent the formation of actin aggregates and achieve a dynamic steady state for myosin motion at a low concentration of gelsolin, we have opted to maintain this gelsolin concentration for our investigation. We have included these points in the Results section to provide the reasoning behind the choice of protein concentrations (pp. 9, lines 271-276).

Minor Comments:

1. *The discussion is very thorough but could use some subheadings to differentiate the different topics explored within that section and better direct the reader on what each section is explaining.*

Thank you for this suggestion. We added subheadings to clarify which topics are discussed in each section in the Discussion (pp. 11, line 341; pp. 12, line 365; pp. 13, line 397; pp.14, line 425).

2. *Can you address the changes which may occur for myosin-induced mechanical work, strain, and power when physiological conditions, such as macromolecular crowding and protein-protein interactions, are introduced?*

We appreciate this insightful comment. Given that the macromolecular crowding in physiological conditions may further increase the viscosity of the solution, the contraction of the actomyosin network will be slowed down by the larger fluid drag (Mogilner & Manhart, *Annu. Rev. Fluid Mech.*, **50**, 347 (2018)). Additionally, protein-protein interactions apply more friction force, counteracting the direction of the actomyosin contraction (Shamipour et al., *Cell* **177**, 1463 (2019)). Therefore, we anticipate that the resulting myosin-induced mechanical power will be smaller. However, macromolecular crowding also increases the local protein concentration, potentially influencing the F-actin architecture through bundling (Köhler et al., *PLoS ONE* **3**, e2736 (2008)). It will be an interesting avenue for future work to explore the impact of such physiological conditions by varying the viscosity of the solution using crowding agents or by introducing actin-membrane binding to induce protein-protein interaction-based friction (Colin et al., *PNAS* **120**, e2300416120 (2020)). Moreover, microtubule networks in cell cytoplasm can also interfere the contraction of the actomyosin networks (Dogterom & Koenderink, *Nat. Rev. Mol. Cell Biol.* **20**, 38 (2019)), where investigation of the microtubule-actin composite networks in vitro will provide valuable insights. The present system serves as a versatile platform for quantitatively studying these effects on mechanical work and ATP consumption. We have included this point in the Results section (pp. 14, lines 444-453).

3. *Figure 3, should be “Branched” instead of “Blanched”.*

We have corrected this typo.

4. In line 228, 6 μM cofilin; [Actin]:[Gelsolin] should be corrected to [Actin]:[Cofilin].

We have corrected this typo.

Response to Reviewer #3

The manuscript by Sakamoto and Murrell describes various experiments to probe energy conversion in actin-myosin networks. Specifically, they reconstitute these gels inside oil-in-water droplet using various molecular controls (cross-linkers, gelsolin, cofilin, etc.) to tune architectural properties of the network. By using an NADH-based assay they determine ATP hydrolysis, and the mechanical work is estimated based on imaging data and assumptions.

Understanding how chemical energy that is consumed at the level of individual molecular motors with each hydrolysis-cycle is converted into mechanical work that leads to networks deformations is timely, interesting, and especially determining the role of network architecture in this makes this work relevant to several mechanical processes in cells. However, in its current form my support for publishing this manuscript is limited by my concerns about the methods the authors use to determine the work and power generated in these networks, on which the central claims of this paper rest. I will outline this point, together with several other concerns and questions below.

We are grateful to the reviewer for identifying our work is timely, interesting, and relevant to several mechanical processes in cells. We agree that some arguments were not clarified enough in the original manuscript. We thus made the following revisions:

1. *The authors estimate the work performed by myosin (and the power), by determining changes in the volume of the network as it contracts which is multiplied by an estimate of the stress. I am concerned that this is a very crude estimate of the work, and little or no evidence is provided as to how accurate this estimate could be. A first aspect that is missing is the fraction energy that is stored in volume conserving modes of deformations of the network, which is not discussed in the text (as an extreme case, imagine anchoring the network to a rigid boundary preventing contraction, implying that all energy goes into volume-conserving modes of deformation). This fraction will be sensitive to the nonlinear elasticity of the network and will likely also dependent on network architecture, which means that even if the work performance would not change with architecture the estimate used by the authors could change. A second aspect that is concerning is the simple argument that is used to estimate the stress production by the myosin motors. Here to, I would expect (possibly large) corrections that are sensitive to network architecture. Finally, a third aspect is that not all work*

is stored is likely stored in elastic energy. There are dissipative effects such as plastic reorganization and fracture (Wollrab et al. Journal of Cell Science 132, no. 4 (2019): jcs219717.), which could be significant.

The authors do mention that their estimate is a bound, but it is unclear how tight the bound is and is unclear how the tightness of the bound varies with conditions, which makes it difficult to use it to compare systems under different conditions. Unfortunately, no direct experimental evidence is given to show how accurate the work estimate is.

Given these issues, I am concerned about presenting these crude estimates in main figures of the paper, and I am concerned that the work and power estimates are used as central observables and arguments that lead to the main conclusions of the paper.

We greatly appreciate these insightful comments, which have provided valuable clarification regarding possible additional contributions to the mechanical work performed by myosin. We agree that the original manuscript did not adequately discuss the limitations of the mechanical work estimation method we employed. In the following, we discuss the potential contributions to mechanical work that were not considered in our estimates.

In our estimation, we used the network volume change as a first-order readout of the mechanical work performed by myosin-generated stress. We referred to the energy stored in the volume variation mode of deformation as $E_{\Delta V}$. However, as the reviewer pointed out, there are other possible contributions to the energy cost that exist during the contraction of the network. In this case, the total energy cost of deformation is expressed as $E_{\text{total}} = E_{\Delta V} + E_{\Delta V=0} + E_{\text{dissipative}} + E_{\text{load/architecture}}$, where $E_{\Delta V=0}$ represents the energy stored in volume conservation modes of deformation (e.g., filament bending), $E_{\text{dissipative}}$ is the dissipated energy during contraction (e.g., filament stretching and breakage), and $E_{\text{load/architecture}}$ is the contribution of load-dependent/architecture-specific stress generated by myosin motors. In the following, we discuss how these missing energies contribute to our estimates of mechanical work.

First, fraction of energy could be stored in volume conservation modes of deformation, $E_{\Delta V=0}$, such as filament buckling. To evaluate the contribution of volume conservation modes, we estimate the buckling energy of F-actin. With a bending stiffness of $k = 7.3 \times 10^{-26}$ N m² (Gittes et al., *J. Cell Biol.* **120**, 923 (1993)), and a filament length $L = 1$ μm , the corresponding energy to buckle is $E_b = kL/2r_c^2 = 4.1 \times 10^{-19}$ J, where $r_c = 300$ nm is the radius of curvature of buckled filaments measured in the contracting actomyosin network (Murrell & Gardel, *PNAS*, **109**, 20820 (2012)). To estimate the maximum contribution of the filament buckling, we assume that the total 6 μM actin (in our experimental condition) forms $L = 1$ μm F-actin and buckles. Since 1 μm F-actin consists of ~ 370 monomers (Kasza et al., *Biophys. J.* **99**, 1091 (2010)), 6 μM actin in a droplet with radius $R = 20$ μm is equivalent to a total number of 3.2×10^5 F-actin with $L = 1$ μm . Assuming all the F-actin is buckled within 10 min during the contraction of the network, the maximum energy used to buckle the filaments per unit time

is $\sim 2 \times 10^{-16} \text{ J s}^{-1}$. This value is two orders of magnitude smaller than the maximum power calculated solely based on the volume variation mode in the main text ($\sim 1.5 \times 10^{-14} \text{ J s}^{-1}$). Thus, the energy used for the buckling of F-actin is not significant compared to the volume variation modes of mechanical power. Although the exact contribution of crosslinking and bundling to the volume conservation modes is not fully understood, we expect that these effects will stiffen the F-actin, making filaments less likely to buckle, and the contribution of architecture will not be significant.

Secondly, not all the energy is stored in elastic energy, but dissipative effects such as plastic reorganization and fracture/severing of the network occur ($E_{\text{dissipative}}$) (Wollrab et al. *J. Cell Sci.* **132**, jcs219717 (2019); Michael et al., *PNAS* **109**, 20820 (2012); Jung et al., *ACS Macro Lett.* **5**, 641 (2016)). To evaluate the contribution of the dissipative effects, we estimate the energy used to break/sever the filaments. The tensile stress required to break a filament under tension is 500 pN (Tsuda et al., *PNAS* **93**, 12937 (1996)). To separate the actin monomers from each other with the typical spacing between monomers, 2.7 nm (Hanson et al., *J. Mol. Biol.* **6**, 46 (1963)), corresponds to the energy cost of severing a filament as $\sim 1.35 \times 10^{-18} \text{ J}$. To estimate the maximum energetic cost of severing, we utilized the observation that filament buckling coordinates the severing, where severing predominantly occurs below a curvature radius of buckled filaments $\sim 300 \text{ nm}$ irrespective of the cross-linking density (Murrell & Gardel, *PNAS*, **109**, 20820 (2012)). Assuming a total number of 3.2×10^5 F-actin with $L = 1 \text{ }\mu\text{m}$ is buckled and severed within 10 min during the contraction of the network, the maximum energy used to sever the filaments per unit time leads $\sim 7.2 \times 10^{-16} \text{ J s}^{-1}$. This value is two orders of magnitude smaller than the maximum power calculated solely based on the volume variation mode in the main text ($\sim 1.5 \times 10^{-14} \text{ J s}^{-1}$). Thus, the contribution of dissipative effects due to severing of F-actin is not significant compared to the volume variation modes of mechanical power. When the network is crosslinked, severing could be enhanced 2-3 times (Murrell & Gardel, *PNAS*, **109**, 20820 (2012)), but even so, the energy dissipation due to severing is still less than the volume variation modes. Although the exact contribution of crosslinking and bundling to the dissipative modes is not fully understood, we expect that these effects will stiffen the F-actin, making filaments less likely to be severed, and the contribution of severing will not be significant.

Furthermore, the reviewer highlighted the potential need for corrections in myosin-based stress generation depending on different architectures ($E_{\text{load/architecture}}$). This is a valid concern, as we rely on the duty ratio of unloaded myosin and did not consider the contribution of architectures, such as the polarity of bundles, to myosin force generation. For example, previous experimental and numerical simulation studies have shown that the polarity of actin bundles crosslinked by fascin or fimbrin influences the load-dependent myosin stress generation (Weirich et al., *Biophys. J.* **120**, 1957 (2021)). It was demonstrated that myosin moves persistently in unipolar bundled networks crosslinked by fascin, whereas myosin

motion is confined under mixed polar bundled networks crosslinked by fimbrin. This is because myosin binding to opposing F-actin in the bundle resists the myosin movement, resulting in the buildup of a force dipole. Using numerical simulations considering the load-dependence of myosin force generation and polarity of bundles, it was shown that the forces applied to the bundled networks vary by ~4 times at maximum depending on the polarity of bundles for myosin thick filaments composed of 250 heads. In our estimates of mechanical work, we have shown that unipolar bundles produce ~10 times larger mechanical power compared to mixed polar bundles ($\sim 1.2 \times 10^{-14}$ J s⁻¹ in unipolar; $\sim 1.2 \times 10^{-15}$ J s⁻¹ in mixed polar). Therefore, even if we consider the correction of 4 times larger myosin stress in mixed polar bundles, the mechanical power of unipolar bundles is still larger. Hence, the contribution of load/architecture-dependent myosin force generation does not alter the conclusion of our manuscript.

For branched architectures, it was experimentally and numerically shown that larger branching reduces stress propagation due to the inhibition of myosin motion, where a highly branched actin network with many barbed ends pointing outward from the mother filaments could make it difficult for myosin thick filaments to walk or slide (Muresan et al., *Nat. Commun.* **13**, 7008 (2022)). In this case, the effective stress applied to myosin is decreased, thus the correction to the stress leads to lower mechanical power. In the main text, we argued that the mechanical power generated within Arp2/3 branched network is reduced compared to control and linear F-actin architecture; thus, the correction to the branched architecture further signifies the influence of branching in our estimates of mechanical power. Overall, the contribution of load/architecture-dependent myosin stress generation will not change the main conclusion of this work.

In summary, our estimates of myosin-based mechanical work do not encompass: 1. volume conservation modes, 2. dissipative effects, 3. load/architecture-dependent myosin force generation. As a result, our estimates provide a lower bound; yet, considering these additional contributions will not alter the main conclusion. We have incorporated these considerations into the main text to clarify the assumptions and limitations in our mechanical work estimates (**pp. 6, lines 145-151, Supplementary Note 1**). Quantifying these unconsidered contributions in experiments remains a challenge, and future works, including agent-based simulations, could contribute to characterizing energy allocation in different deformation modes and exploring load-dependence and dissipative effects.

2. An NADH assay is used to estimate energy consumption. One concern I have with this approach is that not all ATP hydrolysis by myosin necessarily results in force-generation in the network, and this might also depend on network architecture. How the authors performed controls to investigate how much of the ATP consumption results can be effectively transferred to the actin network?

Thank you for this comment. Firstly, we agree that not all ATP hydrolysis by myosin contributes to force generation. However, our first objective is to understand how F-actin architecture alters the ATP consumption rates of myosin. For this aim, measuring architecture-dependent ATP consumption using the NADH assay is an appropriate method.

On the other hand, when investigating how much of the ATP consumption can be effectively transferred to the actin network, we need to make an approximate assumption on energy conservation. Here, we assume that the conversion of chemical energy input per unit time, $\Delta G_{\text{ATP}}/\Delta t$, into mechanical power, P_{max} , is described as $\Delta G_{\text{ATP}}/\Delta t = P_{\text{max}} + E_{\text{diss}}$, where we denote the chemical energy not used for mechanical work as E_{diss} . To estimate the energy conversion efficiency, η , indicating how much of the ATP hydrolysis energy is effectively converted into mechanical work, we calculate $\eta = P_{\text{max}}/(\Delta G_{\text{ATP}}/\Delta t)$. We have added an additional data for the estimation of energy conversion efficiency (**Supplementary Fig. 8**). The efficiency increases with higher myosin concentration, might be because the elevated myosin concentration may increase the probability of temporary connections within the actin network via myosin thick filaments, improving the transmission of contractile forces (Jung et al., *Comp. Prat. Mech.* **2**, 317 (2015); Jung. et al., *Cytoskeleton (Hoboken)* **76**, 517 (2019)) (**Supplementary Fig. 8A**). In fimbrin-crosslinked mixed polar bundles, the efficiency is lower than the other crosslinked networks, attributed to significantly lower power (**Supplementary Fig. 8B**). The Arp2/3-nucleated branched network exhibits low energy conversion efficiency, while the linear network nucleated by mDia1 enhances efficiency when mixed with the branched network (**Supplementary Fig. 8C**). Notably, these values are ~100 times smaller than the maximum efficiency estimated for the contraction of skeletal muscle fiber (~0.36) (He et al., *J. Physiol.* **517**, 839 (1999)), highlighting the importance of the structural organization of the actomyosin system to efficiently extract macro-scale work from the molecular consumption of energy.

Together, these results provide an estimate of how much of the ATP consumption were converted into mechanical work in the present system. As the reviewer suggested in comment 1, it is noteworthy that the estimate of mechanical power in the present study represents a lower bound. Consequently, the estimated energy conversion efficiency is also a lower bound. We mentioned this energy conversion efficiency in the Results section in the main text (**pp. 6, lines 167-173; pp. 8, lines 208-210; pp. 9, lines 255-257**).

Additional Supplementary Figure 8 : The lower bound efficiency of energy conversion. The efficiency of the conversion of the free energy from ATP hydrolysis per unit time ($\Delta G_{\text{ATP}}/\Delta t$) to mechanical power (P_{max}) is estimated through the ratio of the maximum power to ATP consumption rate in a droplet, $\eta = P_{\text{max}}/(\Delta G_{\text{ATP}}/\Delta t)$. The ATP consumption rate is converted to the free energy of ATP hydrolysis by using the literature value of free energy released from ATP hydrolysis in physiological condition $\sim 50\text{-}70$ kJ mol $^{-1}$. It is noteworthy that the estimate of mechanical power in the present study represents a lower bound. Consequently, the estimated energy conversion efficiency is also a lower bound. **(A)** Barplots showing the energy conversion efficiency at different myosin concentration ($n=5$ droplets and $N=3$ independent experiments for 12.5 nM; $n=3$ and $N=5$ for 25 nM; $n=10$ and $N=8$ for 50 nM). The efficiency is larger for the higher myosin concentration, which could be because the higher myosin concentration may increase the probability of transient crosslinking within the actin network via myosin thick filaments, which may transmit contractile forces more efficiently. **(B)** Barplots showing the energy conversion efficiency in various crosslinkers ($n=10$ and $N=8$ in without crosslinkers; $n=8$ and $N=5$ in fascin; $n=5$ and $N=3$ in fimbrin; $n=8$ and $N=5$ in α -actinin). Although fimbrin-crosslinked mixed polar bundles can reduce the ATP consumption, due to significantly small mechanical power, the resulting efficiency is lower than that of the other crosslinked networks. **(C)** Barplots showing the energy conversion efficiency in various nucleator compositions ($n=10$ and $N=8$ in without nucleators; $n=10$ and $N=9$ for Arp2/3 only; $n=6$ and $N=5$ in mDia1 only; $n=9$ and $N=6$ in [Arp2/3]:[mDia1] = 1:1; $n=8$ and $N=4$ in [Arp2/3]:[mDia1] = 1:0.1). The efficiency of energy conversion is low for the Arp2/3-nucleated branched network, while the linear network nucleated by mDia1 improves the efficiency when it is mixed with the branched network. *, **, and *** correspond to $p < 0.05$, $p < 0.01$, and $p < 0.001$, respectively. n.s.: not significant.

3. Fig 1n shows energy input versus mechanical output and the authors argue that this relation is linear. Given the limited number of data points (only 3) and large error bar on the 50 nM case, I think the evidence is thin. More (precise) data would be needed to draw this conclusion.

We thank the reviewer for this comment. The intention behind presenting this plot is not to assert the linearity between energy input and mechanical output, but to provide an overview

of mechanical power and ATP consumption in a phase space. Accordingly, we have revised the sentence without mentioning linearity (**pp. 6, lines 156-157**). While our aim is not to demonstrate linearity in this plot, it's important to note that fitting was performed for each of the 22 individual data points; thus, we believe that the number of points is sufficient for fitting.

4. A separate question tangentially related to this figure: What percentage of the energy input is converted into work, and how does this efficiency depend on network architecture? The line about “how distinct F-actin architectures impact ATP consumption and energy conversion efficiency” made me expect results on this topic.

This is a great suggestion. We have looked into how efficiently energy is converted and how it depends on the structure, as described in response to comment #2. Briefly, the efficiency goes up for the higher myosin concentration. This might be because higher myosin concentration increases the chance of temporary connections in the actin network through myosin thick filaments, making contractile forces transmit more effectively (**Supplementary Fig. 8A**). In mixed polar bundles crosslinked by fimbrin, the efficiency is lower compared to other crosslinked networks because of a lower power (**Supplementary Fig. 8B**). The branched network nucleated by Arp2/3 shows low energy conversion efficiency. However, when mixed with the linear network nucleated by mDia1, the efficiency improves (**Supplementary Fig. 8C**). We have added these additional data in **Supplementary Fig. 8** and discussed it in the Results section (**pp. 6, lines 167-173; pp. 8, lines 208-210; pp. 9, lines 255-257**).

5. The authors present strain fields in various main text figures, but they are not discussed in detail. Can they expand on the presence of both large compressive and extensile strains and the homogeneous structure of these strain fields?

We appreciate this suggestion. In the previous main text, we did not discuss the structure of strain fields. The homogeneous structure of strain fields corresponds to the uniform contraction of the entire network, as observed in spontaneously polymerized networks and fascin-bundled networks. On the other hand, the Arp2/3 network exhibited spatially nonuniform compressive and extensile strain. This heterogeneity arises because the Arp2/3-nucleated flower-like networks diffuse in space without global contraction. In the case of formin-nucleated networks, the contraction of the network forms aster-like structures that merge over time, leading to a local compressive strain field. We have incorporated these descriptions of strain fields into the Results section (**pp. 7, lines 192-193; pp. 8, lines 225-227; pp. 9, lines 241-242**).

6. Some relevant context in the discussion is missing. A relevant pioneering paper is not

mentioned Bendix et al. Biophysical journal 94, no. 8 (2008): 3126-3136. Also, I want to bring to the attention: Nitta et al. Nature Materials 20, no. 8 (2021): 1149-1155 and Jia et al. Nature Materials 21, no. 6 (2022): 703-709. The latter specifically discussed work production in actin/myosin networks.

We thank the reviewer for introducing these pioneering papers. We have included these works in the Discussion section in the main text (**pp. 12, lines 355-358; pp. 15, lines 460-466**), improving the relevance to previous works.

Reviewers' Comments:

Reviewer #1:

Remarks to the Author:

The points I raised during the revisions have been adequately addressed and the authors have done a good job of improving the manuscript. However, I still have some minor concerns about my first point relating to the definition and characterization of the dynamic steady state. Once the authors have responded to these comments, I will be pleased to recommend the paper for publication in Nature Communications.

The authors define the dynamic steady state as the propensity to retain a constant F-actin density and not contract. They also added a quantitative way to estimate if the network is in a dynamic steady state or not (with a macroscopic strain rate equal to 0 and a magnitude of velocity above 0). My questions are the following:

- The molecular origin of this dynamic steady state is still unclear to me. Could the authors clarify this point?
- Another point that is not clear from the manuscript is if the authors consider that the cofilin network is in a dynamic steady state or not. Strain rate and magnitude of velocity should be computed in this case too. Otherwise, Supplementary Figure 12 is very confusing to read since we do not know which networks are fulfilling the criteria of being in a dynamic steady state.
- The authors computed the strain rate and the magnitude of velocity for the gelsolin and the phalloidin cases in presence of myosin. As the definition of the dynamic steady state is given in reference to myosin, it would be nice to compute the strain and velocity in absence of myosin, in order to confirm that the phenomenon the authors observe is due to the myosin and not to thermal fluctuations of small filaments created following the fragmentation of the network by gelsolin.

Reviewer #2:

Remarks to the Author:

The authors have addressed all my comments clearly. I appreciate their thorough work.

Reviewer #3:

Remarks to the Author:

The authors have carefully and thoughtfully addressed the reviewer's comments. As I mentioned in my previous report, this work is timely and interesting. I appreciate the comments and additions the author's made in response to my biggest concern regarding the work and stress estimates. While I still feel that this is an issue that requires more attention in the future, given the state-of-the-art in the field and that the assumptions and caveats are now carefully described in the manuscript, I think this manuscripts can be published in this form in Nature Comm. However, I do want to point out to the authors that a volume-conserving shear mode could also include stretching contributions of actin (not just compression that leads to buckling), and i wouldn't be surprised if this would end up being a rather significant contribution. It would be worthwhile mentioning this in the manuscript.

Response Letter
F-actin architecture determines the conversion of chemical energy into mechanical work
R. Sakamoto & M.P. Murrell

We greatly appreciate all the reviewers for carefully reading our revised manuscript and for their insightful comments. In light of their suggestions, we have made changes to the manuscript and responded to each comment below.

Response to Reviewer #1

The points I raised during the revisions have been adequately addressed and the authors have done a good job of improving the manuscript. However, I still have some minor concerns about my first point relating to the definition and characterization of the dynamic steady state. Once the authors have responded to these comments, I will be pleased to recommend the paper for publication in Nature Communications.

We are grateful for the positive feedback on our revised manuscript. We acknowledge the concerns raised regarding the definition and characterization of the dynamic steady state. In response, we have made the following revisions.

The authors define the dynamic steady state as the propensity to retain a constant F-actin density and not contract. They also added a quantitative way to estimate if the network is in a dynamic steady state or not (with a macroscopic strain rate equal to 0 and a magnitude of velocity above 0). My questions are the following:

1. The molecular origin of this dynamic steady state is still unclear to me. Could the authors clarify this point?

We appreciate the reviewer's comment. The molecular origin of the dynamic steady-state lies in the nature of skeletal muscle myosin II, which is known for its fast motor activity and low duty ratio (~0.05 (Uyeda et al., *J. Mol. Biol.* (1990))), resulting in rapid binding and unbinding cycles to F-actin. This rapid cycling induces significant fluctuations of F-actin perpendicular to the filament axis without causing macroscopic contraction, named as 'plucking' in our previous work (Seara et al., *Nat. Commun.* (2018)). These reversible F-actin plucking events arise from transient and diverse interactions between non-aligned myosin and F-actin, and they occur regardless of myosin isoforms (Seara et al., *Nat. Commun.* (2018)). The plucking mode of myosin contraction has been theoretically studied and well-established (Lenz, *Phys. Rev. X* (2014); Ronceray et al., *Soft Matter* (2019)). We have clarified this point by including the possible molecular origin of the fluctuation in the Results section (**pp. 10, lines 282-288**).

2. Another point that is not clear from the manuscript is if the authors consider that the cofilin network is in a dynamic steady state or not. Strain rate and magnitude of velocity should be computed in this case too. Otherwise, Supplementary Figure 12 is very confusing to read since we do not know which networks are fulfilling the criteria of being in a dynamic steady state.

We appreciate this suggestion. We added an additional analysis on the cofilin-severed

network (**Revised Supplementary Fig. 11**). We found that the cofilin-severed network exhibits a finite total strain due to the initial contraction and the formation of actin aggregates, suggesting that cofilin-mediated severing is insufficient to achieve a dynamic steady-state under the present experimental conditions. We have included this point in the Results section (pp. 10, lines 277-280).

Revised Supplementary Figure 11. The actomyosin system severed by gelsolin is in a dynamic steady-state. (A) PIV analysis of actomyosin network contraction in the presence of 56 nM gelsolin or 6 μM cofilin, comparing with and without myosin, and 17 μM phalloidin with myosin. Yellow vectors represent total displacement \mathbf{u} over 20 min, with vector magnitudes normalized by the maximum displacement (top). The colormap represents local strain fields (bottom). (B) Macroscopic strain rate of the actin network ($\dot{\epsilon} \equiv -(\nabla \cdot \mathbf{v})$) over time calculated from the instantaneous velocity field \mathbf{v} between each frame ($n=9$ droplets and $N=8$ independent experiments in gelsolin with myosin (+) and $n=8$ and $N=3$ in gelsolin without myosin (-); $n=9$ and $N=6$ in cofilin with myosin (+) and $n=6$ and $N=3$ in cofilin without myosin (-); $n=8$ and $N=5$ in phalloidin). (C) Boxplots showing the total strain calculated from the integration of the strain rate in (B) over 20 min ($\int \dot{\epsilon} dt$). The total strain is approximately 0 in the gelsolin-severed network, indicating a dynamic steady-state. By contrast, the cofilin-severed network exhibits a finite total strain due to the initial contraction and the formation of actin aggregates, suggesting that cofilin-mediated severing is insufficient to achieve a dynamic steady-state under the present experimental conditions. (D) Spatially averaged magnitude of velocity ($\langle |\mathbf{v}| \rangle$) over time ($n=9$ droplets

and N=8 independent experiments in gelsolin with myosin (+) and n=8 and N=3 in gelsolin without myosin (-); n=9 and N=6 in cofilin with myosin (+) and n=6 and N=3 in cofilin without myosin (-); n=8 and N=5 in phalloidin). (E) Boxplots showing the maximum averaged magnitude of velocity ($\langle |v| \rangle_{\max}$) extracted from (D). The observed velocity in the presence of myosin is larger than that of without myosin, indicating that myosin activity induces larger fluctuations in F-actin compared to thermal fluctuations. ** and *** represent $p < 0.01$, and $p < 0.001$, respectively. Scale bars, 10 μm .

3. *The authors computed the strain rate and the magnitude of velocity for the gelsolin and the phalloidin cases in presence of myosin. As the definition of the dynamic steady state is given in reference to myosin, it would be nice to compute the strain and velocity in absence of myosin, in order to confirm that the phenomenon the authors observe is due to the myosin and not to thermal fluctuations of small filaments created following the fragmentation of the network by gelsolin.*

We are thankful for this insightful suggestion. We computed the strain and velocity in the absence of myosin (**Revised Supplementary Fig. 11**). We found that the observed velocity in the presence of myosin is significantly larger than that of without myosin, indicating that myosin activity induces larger fluctuations in F-actin compared to thermal fluctuations. This point is included in the caption of **Revised Supplementary Fig. 11**.

Response to Reviewer #2

The authors have addressed all my comments clearly. I appreciate their thorough work.

We greatly appreciate the reviewer's comment.

Response to Reviewer #3

The authors have carefully and thoughtfully addressed the reviewer's comments. As I mentioned in my previous report, this work is timely and interesting. I appreciate the comments and additions the author's made in response to my biggest concern regarding the work and stress estimates. While I still feel that this is an issue that requires more attention in the future, given the state-of-the art in the field and that the assumptions and caveats are now carefully described in the manuscript, I think this manuscripts can be published in this form in Nature Comm. However, I do want to point out to the authors that a volume-conserving shear mode could also include stretching contributions of actin (not just compression that leads to buckling), and i wouldn't be surprised if this would end up being a rather significant contribution. It would be worthwhile mentioning this in the manuscript.

We appreciate the reviewer for evaluating our revision was careful and thoughtful, as well as the insightful additional notion on the stretching contribution of actin. Following the

suggestion, we estimated the contribution of the elastic energy stored in the F-actin by stretching. To estimate the maximum contribution of the filament stretching, we assume that the total 6 μM actin (in our experimental condition) forms $L = 1 \mu\text{m}$ F-actin, in which they are stretched until breakage by reaching the critical tensile stress of filament breaking $F_c \sim 500$ pN (Tsuda et al., *PNAS* (1996)). We consider the enthalpic stretching regime as the network is strongly contracted, in which filaments behave similar to a rigid rod (linear spring), respond to longitudinal tensile stresses by increasing their contour length (Lenz, *Phys. Rev. X* (2014); Broedersz et al., *Rev. Mod. Phys.* (2014); de la Cruz & Gardel, *J. Biol. Chem.* (2015)). The extension of the F-actin with $L = 1 \mu\text{m}$ under tensile stress of $F_c \sim 500$ pN is estimated to be $\Delta L = F_c/K \sim 11$ nm, using a value of the stiffness $K = 43.7 \times 10^{-3}$ N/m of a single actin filament with $L = 1 \mu\text{m}$ (Kojima et al, *PNAS* (1994)). It should be noted that a complete model considering both entropic and enthalpic stretching previously estimated $\Delta L \sim 4$ nm at tensile stress of ~ 500 pN (de la Cruz & Gardel, *J. Biol. Chem.* (2015)). Using the stiffness $K = 43.7 \times 10^{-3}$ N/m (Kojima et al, *PNAS* (1994)) and the extension length $\Delta L = 11$ nm, enthalpic stretching energy of F-actin yields $K\Delta L^2/2 \sim 2.6 \times 10^{-18}$ J. Using the total number of 3.2×10^5 F-actin with $L = 1 \mu\text{m}$ in a droplet with radius $R = 20 \mu\text{m}$, the total maximum energy stored in the stretching mode per unit time during 10 min of contraction is estimated to be $\sim 1.4 \times 10^{-15}$ J s $^{-1}$. This value is one order of magnitude smaller than the maximum power calculated solely based on the volume variation mode in the main text ($\sim 1.5 \times 10^{-14}$ J s $^{-1}$). Thus, the energy used for the stretching of F-actin is not significant compared to the volume variation modes of mechanical power. We note that the presence of the critical forces of filament breakage limits the contribution of the energy stored in the stretching mode, in which torsional strain imposed on the F-actin during contraction may decrease the critical breakage forces (Tsuda et al., *PNAS* (1996)), thereby the contribution of the stretching mode could be further reduced.

We have incorporated this discussion on the possible additional contribution of the stretching of actin, which further clarified the assumptions and limitations of the mechanical power estimates (pp. 6, lines 148, Supplementary Note 1).

Reviewers' Comments:

Reviewer #1:

Remarks to the Author:

The authors have now responded to all my comments. I recommend publication of this manuscript in Nature communications.

Reviewer #3:

Remarks to the Author:

I really appreciate the detailed answer to the final comment on the stretching contribution the the energy. I recommend publication of the manuscript in this form.